# The LipoGlo reporter system for sensitive and specific monitoring of atherogenic lipoproteins

James H. Thierer [1,2], Stephen C. Ekker [3] & Steven A. Farber [1,2]

Apolipoprotein-B (ApoB) is the structural component of atherogenic lipoproteins, lipid-rich particles that drive atherosclerosis by accumulating in the vascular wall. As atherosclerotic cardiovascular disease is the leading cause of death worldwide, there is an urgent need to develop new strategies to prevent lipoproteins from causing vascular damage. Here we report the LipoGlo system, which uses a luciferase enzyme (NanoLuc) fused to ApoB to monitor several key determinants of lipoprotein atherogenicity including particle abundance, size, and localization. Using LipoGlo, we comprehensively characterize the lipoprotein profile of individual larval zebrafish and collect images of atherogenic lipoprotein localization in an intact organism. We report multiple extravascular lipoprotein localization patterns, as well as identify Pla2g12b as a potent regulator of lipoprotein size. ApoB-fusion proteins thus represent a sensitive and specific approach to study atherogenic lipoproteins and their genetic and small molecule modifiers.

[1] Carnegie Institution for Science Department of Embryology, 3520 San Martin Drive, Baltimore, MD 21218, USA. [2] Johns Hopkins University Department of Biology, 3400N Charles Street, Baltimore, MD 21218, USA. [3] Department of Biochemistry and Molecular Biology, Mayo Clinic, 200 First Street SW, Rochester, MN 55905, USA. Correspondence and requests for materials should be addressed to S.A.F. (email: Farber@Carnegiescience.edu)

Apolipoprotein B-containing lipoproteins (ApoB-LPs) are the etiological agents of atherosclerotic cardiovascular disease[1], which is the leading cause of mortality worldwide[2]. ApoB-LPs serve to shuttle lipids throughout the circulation, but occasionally cross the vascular endothelium to form lipid-rich deposits within the vascular wall that develop into atherosclerotic plaques[1]. ApoB-LPs are frequently characterized indirectly through measurement of their triglyceride and cholesterol contents, and lipid-lowering therapies such as statins[3] are used to reduce cardiovascular disease risk by lowering the levels of cholesterol carried by atherogenic lipoproteins (often called bad cholesterol).

Indirect characterization of lipoproteins with lipid measurements, however, provide very limited information on properties such as the concentration and size distribution of ApoB-LPs, both of which are key determinants of atherogenic potential. For example, serum ApoB levels directly reflect the concentration of ApoB-LP particles and show a stronger correlation with cardiovascular disease risk than lipid metrics[4,5]. The size distribution of lipoprotein particles is also relevant to cardiovascular disease risk, as there are numerous classes of ApoB-LPs that can be differentiated by size and show varying degrees of atherogenicity[6]. Low-density lipoproteins (LDLs) are the smallest and most abundant class of ApoB-LPs and are thought to be the primary drivers of atherosclerosis[1]. There is significant size variation within the LDL particle class, and smaller LDL particles are associated with increased atherogenicity[7]. For example, approximately 25% of the adult population produces unnaturally small LDL particles, and as a result have ~3-fold higher risk for cardiovascular disease[8]. The higher atherogenic potential of small dense LDL particles has been attributed to a combination of three properties[7], including increased rates of intimal invasion, reduced receptor-mediated clearance, and increased susceptibility to oxidation.

Many of the genetic and environmental factors governing ApoB-LP size and abundance remain undiscovered or poorly characterized[9–11], and even fewer have been successfully targeted pharmaceutically[12–14]. It has proven particularly difficult to identify drugs that modulate ApoB-LP size and abundance because the simplified model systems typically used in high-throughput drug screening (such as cultured cells or invertebrate models) do not recapitulate the complex multi-organ physiology responsible for ApoB-LP homeostasis. While lipoproteins are studied extensively in mammalian models[15], these systems are not conducive to high-throughput drug discovery. By contrast, the larval zebrafish model system has proven to be a powerful system for in vivo drug discovery[16], as it recapitulates all major aspects of vertebrate physiology in a small, transparent, rapidly developing organism. However, existing assays are not sensitive enough to characterize ApoB-LPs in individual larval zebrafish[17–19], each of which contains only a few nanoliters of plasma.

Here we describe LipoGlo, a sensitive and specific reporter of atherogenic lipoproteins. Modern genome engineering techniques are used to fuse the endogenous apoB gene in zebrafish with an engineered luciferase reporter (NanoLuc), such that each atherogenic lipoprotein is tagged with a light-emitting molecule. NanoLuc is an optimized luciferase reporter that generates a quantitative chemiluminescent signal through processing of its substrate molecule, furimazine[20]. This reporter is remarkably bright (~100 times brighter than firefly luciferase), small (19.1 kDa), stable, and provides robust signal-to-noise ratios that enable accurate detection even at femtomolar concentrations[20]. Using this reporter, we develop several independent assays to characterize distinct aspects of the ApoB-LP profile. These include a plate-based assay to measure lipoprotein quantity (LipoGlo counting), a gel-based assay to measure lipoprotein size (LipoGlo electrophoresis), and chemiluminescent imaging to visualize lipoprotein localization (LipoGlo microscopy).

We also perform extensive validation of these assays in vivo by showing conserved responses to genetic, pharmacological, and dietary manipulations in live zebrafish larvae (summarized in Fig. 1b). Finally, we leverage the discovery potential of these assays to identify unexpected patterns of ApoB-LP localization[21], as well as identify the poorly characterized gene pla2g12b[22] as a potent regulator of lipoprotein particle size that is conserved across vertebrates.

We deploy LipoGlo in larval zebrafish as this organism is uniquely well-suited for high-throughput genetic and small-molecule screening, as well as whole-organism imaging. However, LipoGlo represents a highly generalizable tool that can be expanded to function in additional model systems, and customized with different reporters depending on the research question. This technique has the potential to transform our understanding of atherogenic lipoprotein biology, which may have important clinical repercussions in the treatment of atherosclerotic cardiovascular disease.

## Results

**Engineering the LipoGlo reporter.** Coupling lipoproteins to a light-emitting reporter (LipoGlo) facilitates characterization of several distinct aspects of the lipoprotein profile (Fig. 1a). A single copy of the ApoB protein is present on each ApoB-LP, making it an ideal scaffold for creating a reporter of ApoB-LPs[23,24] (Fig. 1b). In mammals there is a single APOB gene that can be post-transcriptionally edited into two isoforms: the full-length APOB-100 expressed primarily in the liver, and the truncated APOB-48 isoform expressed in the intestine[25,26]. Although the zebrafish genome contains three paralogs of APOB, a single paralog (apoBb.1) is the dominant isoform, accounting for approximately 95% of the apoB messenger RNA (mRNA) and protein in larval zebrafish[27]. Known functional elements of ApoB are well conserved in zebrafish, including both the microsomal triglyceride transfer protein (Mtp) interacting[28] and LDL-receptor binding[29] domains (Supplementary Fig. 1a). However, the APOB-48 editing site required for production of the truncated (intestine-specific) version of APOB[26] appears to be completely absent in zebrafish (Supplementary Fig. 1b). Thus, ApoB-LPs produced by the intestine and liver can be simultaneously tagged with a carboxy-terminal fusion to ApoBb.1 in zebrafish.

The NanoLuc coding sequence was introduced as a carboxy-terminal fusion to the endogenous apoBb.1 gene in zebrafish through homology-directed repair of a double-stranded break[30]. Capped mRNA encoding a TALEN pair targeting the apoBb.1 stop codon was co-injected with a donor DNA construct (Supplementary Fig. 2). Injected embryos were raised to adulthood and their progeny were screened for NanoLuc activity and subsequently for error-free integration at the target locus. The resulting tagged lipoproteins were quantified using the Nano-Glo assay (Promega Corp., N1110), which led us to name this system LipoGlo.

Fish homozygous for the LipoGlo reporter are healthy, fertile, and do not display any abnormal morphological or behavioral phenotypes. Additionally, larvae homozygous for the LipoGlo reporter show a two-fold increase in LipoGlo signal relative to their heterozygous siblings (Supplementary Fig. 2c). Western blotting was used to validate that NanoLuc remains attached to ApoB in vivo (Supplementary Fig. 2d). Together, these data indicate that the LipoGlo reporter signal is directly proportional to ApoB levels.

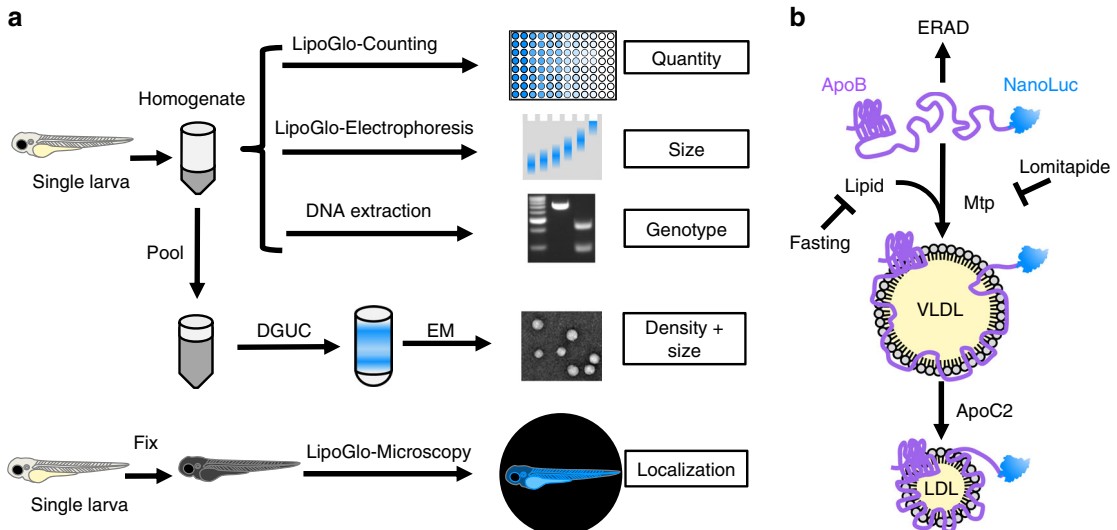

**Fig. 1** Overview of LipoGlo assays and experimental manipulations. **a** Individual larvae carrying the apolipoprotein B (ApoB)-NanoLuc reporter are first homogenized in ApoB-containing lipoprotein (ApoB-LP) stabilization buffer. Homogenate can be used for LipoGlo counting (a plate-based assay for NanoLuc activity to measure the total number of ApoB-LPs), LipoGlo electrophoresis (a Native-polyacrylamide gel electrophoresis (N-PAGE) assay to determine the ApoB-LP size/subclass distribution), and DNA extraction for genotyping. Alternatively, lipoprotein density and size can be determined by density gradient ultracentrifugation (DGUC) followed by electron microscopy. To determine localization of ApoB-LPs in situ, individual larvae are fixed in 4% paraformaldehyde (PFA) and mounted in low-melt agarose for chemiluminescent imaging (LipoGlo microscopy). **b** ApoB protein fused to NanoLuc is loaded with lipid through the activity of microsomal triglyceride transfer protein (Mtp) to form very-low-density lipoprotein (VLDL) particles. In the absence of lipidation, the protein will be rapidly degraded by endoplasmic-reticulum-associated protein degradation (ERAD). VLDL is lipolyzed by serum lipases that use Apoc2 as an obligate cofactor to produce smaller lipoprotein classes such as LDL. Here we investigate the effects of (i) genetic manipulations (mutations in *mtp* and *apoc2*), (ii) dietary variation (fasting and feeding), and (iii) pharmacological treatment (inhibition of Mtp with lomitapide) on various aspects of the ApoB-LP profile

### LipoGlo counting reveals changes in ApoB-LP abundance.

The LipoGlo-counting method uses a 96-well plate-based assay to detect NanoLuc activity and quantify ApoB-LP abundance. Several genetic, pharmacological, and dietary manipulations were performed to validate that canonical aspects of lipoprotein homeostasis are conserved in zebrafish (Fig. 1b). Individual larvae carrying the LipoGlo reporter were homogenized in a standard volume of ApoB-LP stabilization buffer (100 μL) using either a pellet-pestle or a microplate-horn sonicator. A portion of the homogenate (40 μL) was mixed with an equal volume of Nano-Glo assay buffer and quantified in a plate reader. The remaining homogenate was either stored frozen for later use or used for additional assays (Fig. 1a).

ApoB-LP levels were measured throughout development from 1 to 6 days post fertilization (dpf) using zebrafish carrying the LipoGlo reporter in the wild-type (WT) genetic background (Fig. 2a). During this window of development, embryos are in the lecithotropic (yolk-metabolizing) stage[31]. All nutrients required for development are provided by the maternally deposited yolk, until the yolk becomes depleted between 5 and 6 dpf and the larvae begin to rely on exogenous food. Yolk lipid is packaged into ApoB-LPs by the yolk-syncytial layer (YSL), a specialized embryonic organ that expresses many genes involved in ApoB-LP production including *apoBb.1*[27]. Accordingly, ApoB-LP levels are quite low early in development, but increase between 1 and 3 dpf as more yolk lipid is packaged into ApoB-LPs (Fig. 2a). As the larvae are not provided with food, ApoB-LP levels drop later in development as rates of lipoprotein metabolism and turnover exceed rates of production following yolk depletion.

LipoGlo reporter fish were then crossed with fish harboring mutations in essential components of the ApoB-LP production and breakdown pathways. Mtp is responsible for loading nascent ApoB with lipid to form ApoB-LPs[32], and apolipoprotein-C2 (ApoC2) is a cofactor for lipoprotein lipolysis[33] (outlined in Fig. 1b). As expected, $mtp^{-/-}$ mutants[34] exhibit profound defects in ApoB-LP production detectable from the earliest stages of development (Fig. 2b). By contrast, $apoC2^{-/-}$ mutants[17] produce lipoproteins normally but show significantly reduced levels of particle breakdown and turnover compared to sibling controls (Fig. 2c).

To probe the effects of transient Mtp inhibition on larval lipoprotein homeostasis, larvae were exposed to lomitapide. Lomitapide is a pharmaceutical inhibitor of Mtp used to treat familial hypercholesterolemia in humans[35]. Larvae were treated with 10 μM lomitapide or vehicle control for 48 h (3–5 dpf), and treated larvae showed a more rapid decline in NanoLuc levels than vehicle-treated controls. This observation is consistent with lomitapide inhibiting ApoB-LP production and leading to an accelerated decline of ApoB-NanoLuc levels (Fig. 2d).

To test the effect of food intake on ApoB-LP levels, larvae were subjected to a fasting and re-feeding experimental paradigm. Larvae were fed a standard diet (Gemma 75, Skretting USA) for 5 days (from 5 to 10 dpf) to adapt to food intake and reach a physiologically relevant baseline level of ApoB-LPs. Following the initial feeding period, larvae were fasted for 48 h (sampled every 12 h), re-fed with a high-fat meal of 5% egg yolk[36], and sampled at various time points after the meal (the chase period). ApoB-NanoLuc levels were stable for the first 12 h of the fast, but declined rapidly for the duration of the fasting period (Fig. 2e, 0–48 h). Following the high-fat meal (6 h of feeding from time point 48 h to time point 54 h), there was an immediate increase in ApoB-NanoLuc levels (Fig. 2e, 48–120 h). ApoB-NanoLuc levels did not recover to their pre-fasted state following the high-fat meal, but rather remained at an intermediate level for a prolonged period (the duration of the chase period, 72 h).

### Lipoprotein size revealed by LipoGlo electrophoresis.

There are numerous classes of ApoB-LPs, many of which can be

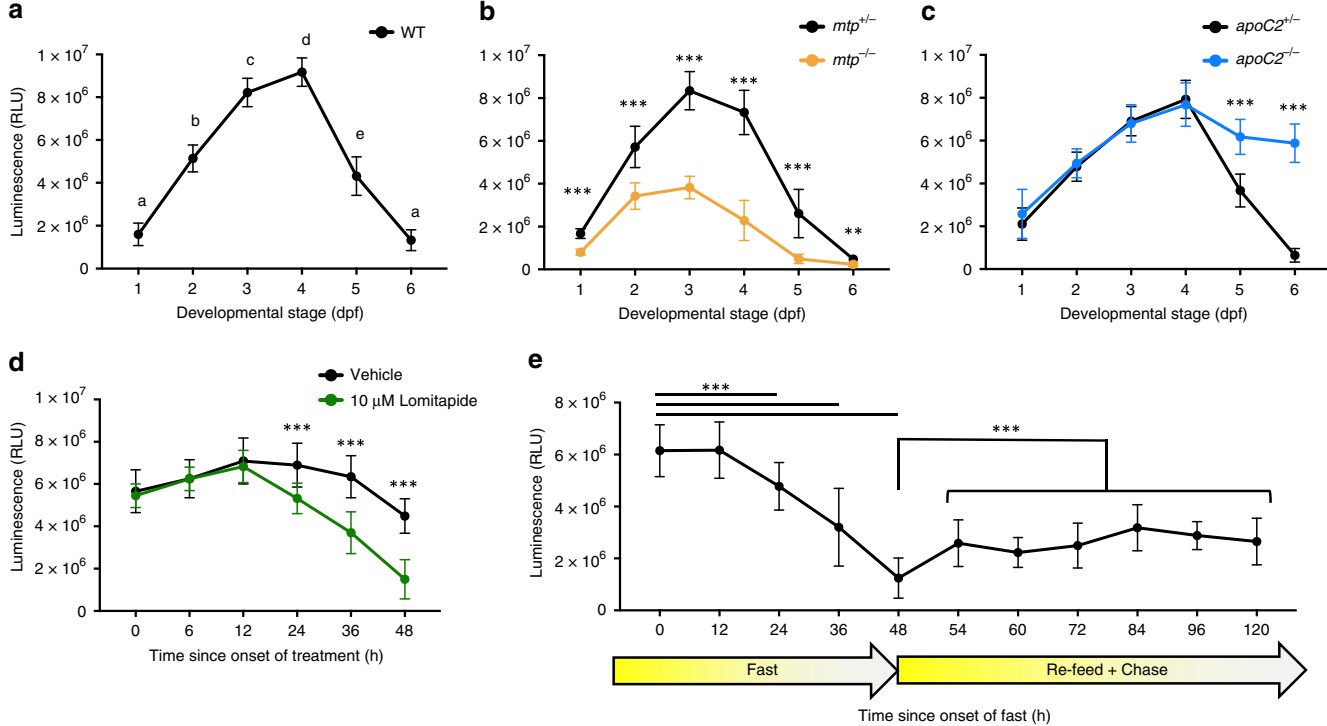

**Fig. 2** LipoGlo counting reveals conserved apolipoprotein B-containing lipoprotein (ApoB-LP) responses to genetic, dietary, and pharmacological stimuli. **a** LipoGlo signal throughout wild-type (WT) larval zebrafish development (1–6 pf). Time points designated with different letters are statistically significantly different by Tukey's honestly significant difference (HSD) with $p < 0.0001$ (degrees of freedom (DF) = 5, $n = 24$, analysis of variance (ANOVA) $p < 0.0001$, Tukey's HSD $p < 0.0001$). **b** Comparison of LipoGlo signal between $mtp^{-/-}$ mutants (defective in lipoprotein synthesis) and $mtp^{+/-}$ siblings during larval development (DF = 11, $n \approx 16$, two-way robust ANOVA $p < 0.0001$ for genotype and stage, Games–Howell $p < 0.001$). **c** Comparison of LipoGlo signal between $apoC2^{-/-}$ mutants (defective in lipoprotein breakdown) and $apoC2^{+/-}$ siblings during larval development (DF = 11, $n \approx 12$, two-way robust ANOVA $p < 0.0001$ for genotype and stage, Games–Howell $p < .0001$). **d** Effect of lomitapide (10 μM, microsomal triglyceride transfer protein (Mtp) inhibitor) on LipoGlo signal (3–5 days post fertilization (dpf)) (DF = 11, $n = 30$, two-way robust ANOVA $p < 0.0001$ for treatment and time, Games–Howell $p < 0.0001$). **e** LipoGlo levels were measured over time throughout a fast, re-feed, and chase period. Larvae were fed a standard diet ad libitum from 5 to 10 dpf, and then were deprived of food for 48 h (fast period). Larvae were then fed a high-fat (5% egg yolk) diet for 6 h, and this meal was chased for 72 h starting at the onset of feeding (48–120 h) (DF = 10, $n = 30$, Welch's ANOVA $p < 0.0001$, Games–Howell $p < 0.0001$). ** denotes $p < 0.001$, and *** denotes $p < 0.0001$. Error bars represent standard deviation. Results represent pooled data from three independent experiments, "$n$" denotes number of samples per data point, mean ± SD shown. Source data are provided as a Source Data file

differentiated based on particle size[37]. Native-polyacrylamide gel electrophoresis (Native-PAGE) has previously been used to separate ApoB-LPs based on size, but is not sensitive enough to detect lipoproteins in individual zebrafish larvae[38]. The LipoGlo electrophoresis method subjects crude larval homogenate to Native-PAGE to separate lipoproteins, followed by an in-gel NanoLuc assay to sensitively detect tagged lipoproteins. To analyze the ApoB-LP size distribution over development and in response to genetic, pharmacological, and dietary manipulations, frozen aliquots of larval homogenate from the experiments outlined above were separated via Native-PAGE (3% gel for 275 Volt-h). Following separation, the glass front plate was removed to expose the gel surface, and an in-gel imaging solution was added to the plate to detect NanoLuc signal. Together, this protocol is referred to as LipoGlo electrophoresis.

Smaller lipoproteins are expected to migrate further into the gel, and larger lipoproteins to show concomitantly less mobility (Fig. 3a). Following electrophoretic separation, ApoB-LPs can be divided into four different classes based on their migration distance. ApoB-LPs that remain within the loading well are classified as the zero mobility (ZM) fraction. Species that do migrate into the gel are classified as either very-low-density lipoproteins (VLDL), intermediate-density lipoproteins (IDLs), or LDLs based on their electrophoretic mobility. DiI-labeled fluorescent LDL (L3482, Thermo Fisher Scientific) is used as a

migration standard to ensure consistent classification of ApoB-LP species between gels, with the migration distance of this species corresponding to 1 ladder unit (LU). Although human DiI-LDL migrates more slowly than NanoLuc-labeled LDL, which is at least partially attributable to migration retardation by DiI (Supplementary Fig. 3g), this band provides a highly reproducible standard for registration and normalization across gels (Supplementary Fig. 3).

In order to define physiologically relevant migration boundaries between ApoB-LP classes, ApoB-LP profiles were compared for WT, $mtp^{-/-}$, and $apoC2^{-/-}$ mutant lines at 4 dpf (Fig. 3a, b). $ApoC2^{-/-}$ mutants are unable to lipolyze VLDL, which allowed us to define the VLDL bin from 0.3 to 1 LUs. Conversely, $mtp^{-/-}$ mutants display a bimodal peak of small LDL-like particles at this stage of development, which was used to define the LDL bin as 1.7–2.4 LUs from the origin. WT larvae have a peak of intermediate-sized lipoproteins at this stage, which corresponds to the IDL region from 1 to 1.7 LUs. ApoB-LPs migrating <0.3 LUs were considered to be in the ZM fraction (Fig. 3a).

Gel images were transformed into plot profiles in ImageJ for quantification (Fig. 3b). The provided Gel Quantification Template (Supplementary Software 1) contains instructions and formulas for automatically calculating bin cutoffs for each ApoB-LP class based on the migration of the DiI standard and quantifying the relative intensity of each bin. Note that LipoGlo

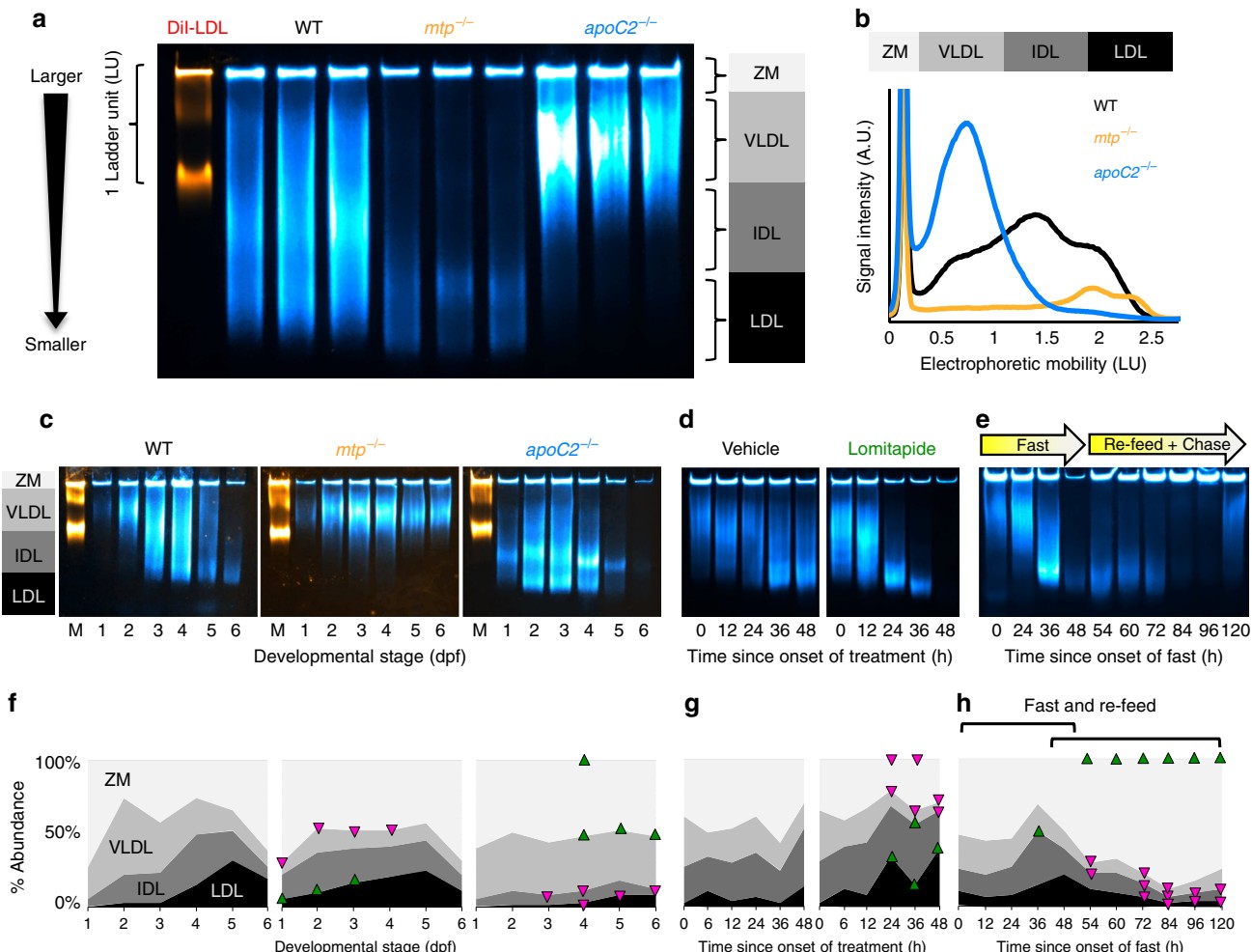

**Fig. 3** Changes in lipoprotein size distribution revealed through coupling Native-polyacrylamide gel electrophoresis (Native-PAGE) to LipoGlo. **a** Representative image of the fluorescent DiI-LDL migration standard and LipoGlo emission from wild-type (WT), $mtp^{-/-}$, and $apoC2^{-/-}$ genotypes (4 days post fertilization (dpf)). ApoB-LPs are divided into four classes based on their mobility, including ZM (zero mobility) and three classes of serum ApoB-LPs (very-low-density lipoproteins (VLDL), intermediate-density lipoproteins (IDL), LDL). Image is a composite of chemiluminescent (LipoGlo, blue) and fluorescent (DiI-LDL, orange) exposures. Gel is a representative image from one of the three independent experiments performed. **b** Vertical plot profile of gel image displayed in **a**, note that the ZM peak has been appended to highlight differences in serum lipoprotein classes. **c–e** Representative LipoGlo PAGE gels (one of three independent experiments shown) and **f–h** quantification of pooled LipoGlo PAGE gel data from larval lysates used in Fig. 2. Note that relative abundance was quantified, so the sum of all species will always equal 100% despite changes in total abundance over time. Relative abundance of subclasses is color coded as shown in **a**. Upward-facing arrowheads (green) indicate significant enrichment of that species at that time point compared to WT, and downward-facing arrowheads (magenta) indicate depletion using the Games–Howell test with a threshold of $p < 0.05$. **f** Subclass abundance at each day of larval development in WT (degrees of freedom (DF) = 5, $n = 9$, Welch's analysis of variance (ANOVA) $p < 0.0001$ for each subclass over time), $mtp^{-/-}$ (DF = 11, $n = 9$, two-way robust ANOVA $p < 0.001$ for VLDL and LDL, Games–Howell $p < 0.01$), and $apoC2^{-/-}$ (DF = 11, $n = 9$, two-way robust ANOVA $p < 0.01$ for all classes, Games–Howell $p < .005$) genetic backgrounds. **g** Subclass abundance from 3 to 5 dpf in larvae treated with 10 μM lomitapide or vehicle control (DF = 11, $n = 9$, two-way robust ANOVA $p < 0.001$ for all classes except IDL, Games–Howell $p < 0.01$). **h** Subclass abundance from 10 to 15 dpf in larvae subjected to a fasting and re-feeding paradigm. The first bracket delineates changes relative to time 0 (the onset of the fasting period), and the second bracket delineates changes relative to time point 48 (the onset of the re-feeding period) (DF = 10, $n = 9$, Welch's ANOVA $p < 0.0001$ for each subclass over time, Games–Howell $p < 0.01$). Supplementary Fig. 4 displays standard deviations for **f–h**. Results represent pooled data from three independent experiments, "$n$" denotes number of samples per data point. Source data are provided as a Source Data file

electrophoresis is only used to determine relative abundance, rather than absolute or total abundance of lipoproteins. To visualize the distribution of ApoB-LP classes over time, each species was color coded with darker colors corresponding to smaller lipoproteins and plotted as an 100% stacked area chart, with the thickness of each shade corresponding to the relative abundance of that species at that time (Fig. 3f–h). Upward-facing green arrowheads or downward-facing magenta arrowheads are used to indicate species that show significant enrichment or

depletion (respectively) relative to the control group (Fig. 3f–h). Additional plots were generated that present the data grouped by ApoB-LP class (rather than genotype) (Supplementary Fig. 4).

LipoGlo electrophoresis experiments over the course of zebrafish larval development revealed that in the early embryonic stages (1–2 dpf), the WT ApoB-LP profile is dominated by VLDL (Fig. 3c, f), which are directly produced by the YSL. By 3 and 4 dpf, VLDL particles have been lipolyzed to generate the smaller IDL and LDL classes. When the maternal yolk has been depleted

(5–6 dpf), and in the absence of exogenous food, VLDL production halts and small lipolyzed LDL particles predominate the lipoprotein profile. By contrast, $mtp^{-/-}$ mutants produce smaller IDL- and LDL-like particles from the earliest stages of development, and $apoC2^{-/-}$ mutants show a VLDL peak that persists throughout larval development (Fig. 3c, f). Pharmacological treatment with lomitapide effectively blocks the production of new VLDL particles (Fig. 3d, g), leading to the accumulation of lipolyzed species such as IDL and LDL.

We observed significant depletion of VLDL and enrichment of LDL following a 48 h fast, reflecting the nutrient dependence of VLDL production. A robust post prandial response was also observed in the distribution of ApoB-LP subclasses of larval zebrafish. A high lipid meal (egg yolk emulsion) produces a significant increase in the ZM band, and progressive depletion of LDL (Fig. 3e, h).

**Concordance between lipoprotein size, density, and mobility.** Electrophoretic mobility in Native-PAGE is a function of both size and charge, so it is important to evaluate whether differences in migration truly reflect different sizes or if they are the result of differentially charged lipoproteins. Density gradient ultracentrifugation (DGUC) is the gold standard for discerning different subclasses of ApoB-LPs, as larger lipoprotein classes are more buoyant resulting from their large lipid core. To evaluate concordance between DGUC and the LipoGlo assays, we developed a DGUC protocol (based on the method described by Yee et al.[39]) to separate pooled larval homogenate into density fractions. We then subjected fractions to (i) LipoGlo electrophoresis to characterize their electrophoretic mobility, (ii) a plate read assay to quantify ApoB-NanoLuc levels, and (iii) negative-staining electron microscopy to visualize particle size directly[40] (Fig. 4). Importantly, denser fractions showed higher electrophoretic mobility and smaller particle sizes across all genotypes, demonstrating that electrophoretic mobility is a reliable method for differentiating ApoB-LP classes and can be used as a proxy to estimate particle size and density.

**Visualization of lipoproteins with LipoGlo microscopy.** The transparency of larval zebrafish offers the unique opportunity to perform whole-mount imaging, which has enabled us to characterize changes in ApoB-LP localization throughout an intact organism. The same developmental, genetic, dietary, and pharmacological manipulations described above (Figs. 2 and 3) were performed, but rather than being homogenized, larvae were fixed in paraformaldehyde (PFA) and mounted in low-melt agarose[41] supplemented with Nano-Glo substrate solution. Mounted larvae were imaged in a dark room on a Zeiss Axiozoom V16 equipped with a Zeiss AxioCam MRm set to collect a single brightfield exposure followed by multiple exposures with no illumination (chemiluminescent imaging).

The differences between WT, $mtp^{-/-}$, and $apoC2^{-/-}$ mutants were most apparent at 6 dpf (Fig. 5a). At this stage, the yolk is depleted and larvae are in a fasted state as no exogenous food has been provided. In WT larvae, signal is quite low throughout the body, but is clearly visible in the lipoprotein-producing tissues (liver and intestine). We observed a previously undescribed association of ApoB with the spinal cord (SC) (Fig. 5b and Supplementary Fig. 5a) as evidenced by colocalization with the central nervous system marker $Tg(Xla.Tubb2:mApple-CAAX)$. This reporter uses the tubulin β-2 promoter from $Xenopus \ laevis$ to drive a membrane-targeted mApple fluorophore specifically in the central nervous system (CNS). A dorsal view revealed enrichment of NanoLuc signal in specific regions of the brain (Fig. 5c). In $mtp^{-/-}$ mutants, ApoB is very low outside of the lipoprotein-producing tissues, consistent with defects in loading ApoB with lipid to form a secretion-competent ApoB-LP. $ApoC2^{-/-}$ mutants show remarkably high signal throughout the body, consistent with their inability to process and turnover lipoproteins (Fig. 5a).

Images were quantified by creating separate regions of interest (ROI) for the viscera, trunk, and head regions and comparing the relative levels of NanoLuc signal in each of these areas. During development, signal was initially highly enriched in the visceral region, which contains the yolk and YSL, and then gradually increased in the trunk and head regions (Fig. 5d, g). This is consistent with the vectoral transport of lipid from the YSL to the circulatory system and peripheral tissues. The distribution of ApoB between these three regions was not significantly changed in $apoC2^{-/-}$ mutants, whereas $mtp^{-/-}$ mutants showed enrichment in the viscera and depletion in the peripheral tissues at all time points (Fig. 5d, g). Results were also grouped by region to facilitate comparison of each class between genotypes (Supplementary Fig. 4).

**Validation of LipoGlo assays with DiI labeling.** In an effort to validate that the unexpected localization patterns of ApoB-LPs was not an artifact resulting from the introduction of the NanoLuc reporter, we developed two orthogonal approaches to monitor ApoB-LPs in zebrafish using a fluorescent lipophilic dye (DiI).

As a means of visualizing LDL localization in vivo, commercially available human DiI-LDL was injected into the zebrafish bloodstream at 2 dpf and then imaged at various time points throughout development (Supplementary Fig. 5c). Immediately following injection (2 dpf), bright DiI fluorescence was readily detectable throughout the vascular system including the caudal artery (CA), caudal vain plexus, and the intersegmental vessels. Imaging at later time points (4 and 6 dpf) revealed significant accumulation in myosepta (MS) and the SC, closely mirroring the localization pattern observed in LipoGlo microscopy. However, in contrast to the LipoGlo microscopy experiments, significant signal accumulated in bright puncta in the ventral posterior of the trunk, which most likely corresponds to macrophages in the caudal hematopoietic tissue (CHT). This result indicates that human DiI-LDL may be immunogenic, either because not derived from zebrafish or it has become oxidized or aggregated during storage. As a negative control, human DiI-LDL was also injected into the yolk of zebrafish larvae (Supplementary Fig. 5d). Immediately after injection, signal was essentially undetectable outside of the yolk, confirming that it has not reached the vasculature. However, approximately 50% of larvae injected into the yolk accrued significant signal outside of the yolk by 6 dpf, where it appeared to mark similar structures as seen in the previous experiment, although signal in the CHT appeared less pronounced. This observation suggests that DiI injected into the yolk (even in the form of human DiI-LDL) could be transferred to endogenous lipoproteins and secreted. To test whether DiI could be used to monitor endogenous ApoB-LPs, we injected DiI directly into the yolk of zebrafish larvae at 1 dpf (Supplementary Fig. 5e). DiI signal closely mirrored the LipoGlo microscopy experiments throughout development. Importantly, this DiI-labeling paradigm showed clear enrichment in the SC and MS by 6 dpf, validating the findings of the LipoGlo microscopy experiment.

To evaluate whether the LipoGlo reporter disrupted lipoprotein homeostasis, we sought to compare the lipoprotein profiles between WT animals and those homozygous for the LipoGlo reporter. As no alternative methods exist to study the lipoprotein profile in zebrafish larvae, it was not possible to

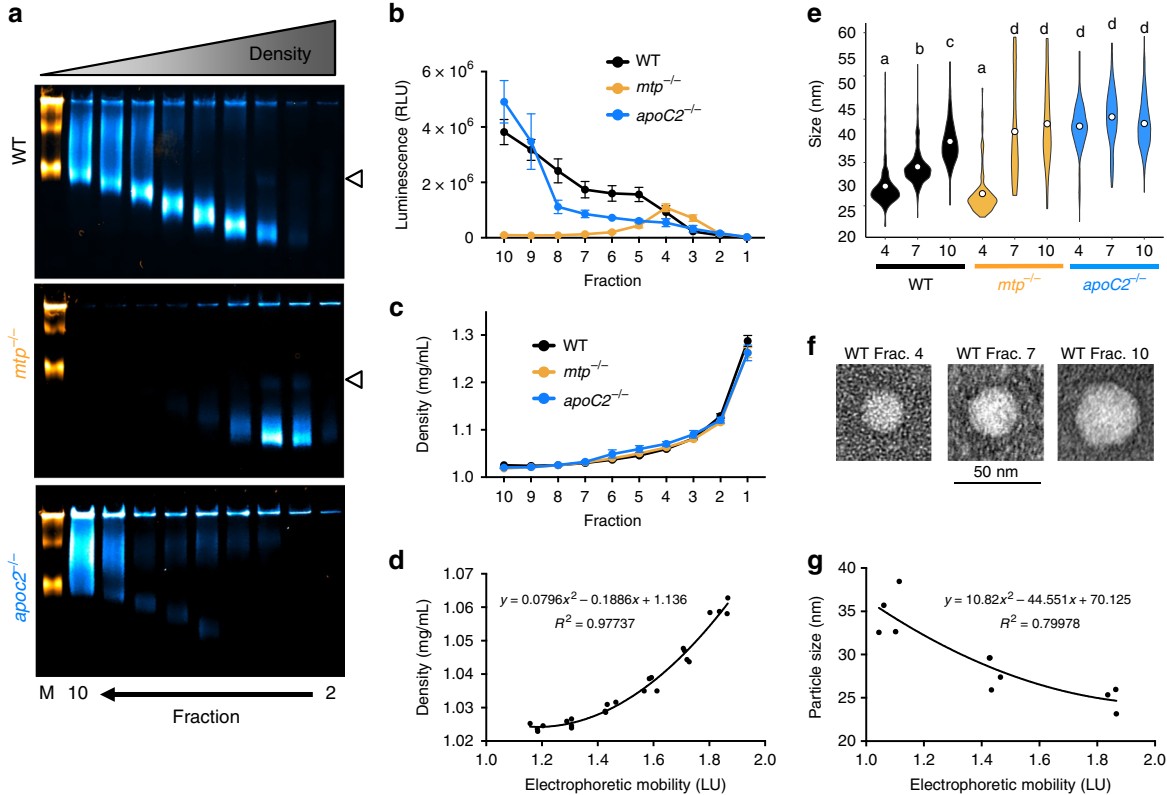

**Fig. 4** Concordance between LipoGlo electrophoresis and classical apolipoprotein B-containing lipoprotein (ApoB-LP) size characterization techniques. Density gradient ultracentrifugation (DGUC) was performed on pooled larval homogenate (4 days post fertilization (dpf)) from wild-type (WT), $mtp^{-/-}$, and $apoC2^{-/-}$, and separated into 10 equal fractions by drip elution (dense bottom fractions eluted first). **a** Fractions 2–10 were subjected to Native-polyacrylamide gel electrophoresis (Native-PAGE), and denser fractions showed higher electrophoretic mobility. Some fractions show a faint lower mobility band (indicated at right by white arrowhead), possibly indicative of lipoprotein dimerization. **b** A plate-based assays of NanoLuc activity revealed the expected enrichment of VLDL in $apoC2^{-/-}$ mutants, and enrichment of LDL in $mtp^{-/-}$ mutants (confirming results reported in Fig. 3b). **c** A refractometer (Bausch and Lomb) was used to determine the refractive index of each fraction and density was calculated via the formula $D = 3.3508 \times RI - 3.4675$. DGUC showed highly reproducible density profiles between replicates and genotypes. **d** The density of WT fractions 4–9 was plotted as a function of peak electrophoretic mobility for that fraction, and the second-order polynomial function ($y = 0.0796 \times 2 - 0.1886x + 1.136$) was able to represent this relationship with remarkable accuracy ($R^2 = 0.97737$), indicating that electrophoretic mobility is a useful proxy for lipoprotein density. **e** Fractions 4, 7, and 10 were subjected to negative-staining electron microscopy to directly visualize the size of particles in each fraction. Letters denote significantly different statistical groups by Games-Howell post hoc test. In the wild-type samples, the average particle diameter was 24.7 ± 5.6, 29.0 ± 4.1, and 34.9 ± 4.7 nm for fractions 4, 7, and 10, respectively. There was no significant difference in particle size between fraction 4 of the WT and $mtp^{-/-}$ mutant samples (average diameter of 23.2 ± 6.6 nm). Particles were nearly undetectable in fractions 7 and 10 in the $mtp^{-/-}$ mutant sample so particle diameter shows enormous variability. ApoB-LPs in each $apoC2^{-/-}$ mutant fraction were significantly larger than all WT fractions, with diameters of 39.0 ± 8.0, 40.9 ± 7.2, and 39.1 ± 5.9 nm respectively (degrees of freedom (DF) = 8, $n \approx 170$, Welch's analysis of variance (ANOVA) $p < 0.0001$, Games–Howell $p < 0.0001$). **f** Representative images of lipoproteins from the three wild-type fractions are shown. **g** The second-order polynomial function $y = 10.82 \times 2 - 44.551x + 70.125$ approximated the relationship between electrophoretic mobility and density in wild-type samples with reasonable accuracy ($R^2 = 0.79978$). Results represent pooled data from four independent experiments. Error bars denote standard deviation. Source data are provided as a Source Data file

make this comparison using larval zebrafish. However, DiI labeling of adult plasma revealed essentially indistinguishable profiles between WT and homozygous LipoGlo adults, although there was significant variation between individuals (Supplementary Fig. 8). These data indicate that the addition of NanoLuc to the carboxy-terminal of ApoB does not disrupt the lipoprotein profile.

**Impact of Pla2g12b on the lipoprotein profile.** In an effort to discover previously uncharacterized regulators of the ApoB-LP profile using LipoGlo, we analyzed a collection of mutants from the zebrafish mutation project[42] that had predicted mutations in genes involved in lipid metabolic pathways. We have discovered that larvae homozygous for an essential splice site mutation

(sa659) in phospholipase A2 group XII B (*pla2g12b*) showed perturbations in their ApoB-LP profile (Fig. 6). Homozygous mutant larvae exhibited lower levels of ApoB at multiple stages (Fig. 6a), and also appeared to have defects in lipoprotein secretion as evidenced by enrichment of visceral ApoB-NanoLuc levels (Fig. 6b–e). However, the most striking defect in *pla2g12b*$^{-/-}$ mutant larvae was a pronounced change in the ApoB-LP size distribution. Even at 1 dpf, significant accumulation of small lipoproteins in the size range of LDL, and depletion of the larger particle classes, were evident (Fig. 6c, d).

## Discussion
Atherogenic lipoproteins are most often isolated from a sample of plasma and quantified using assays for triglyceride and

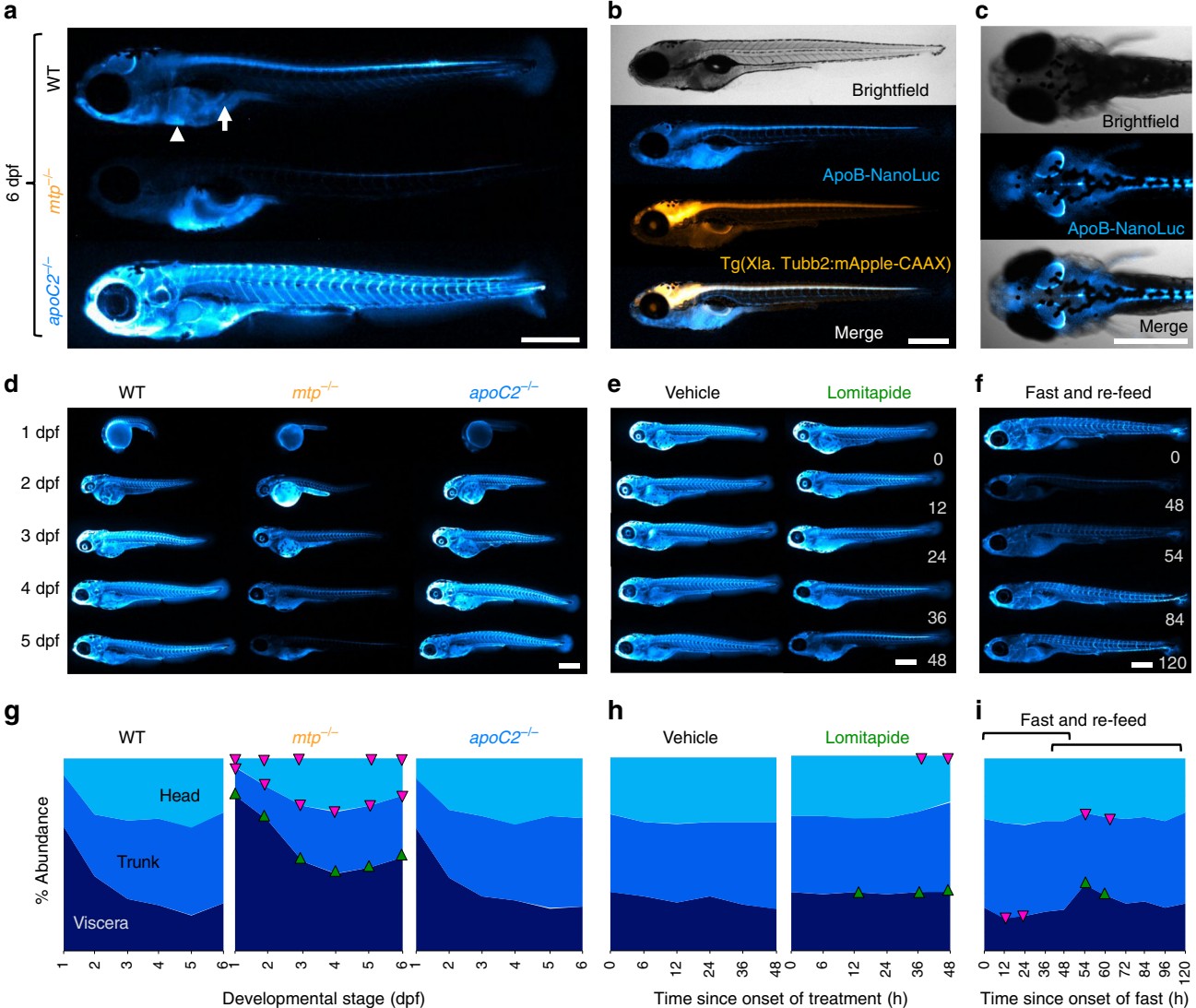

**Fig. 5** Whole-mount imaging of apolipoprotein B-containing lipoprotein (ApoB-LP) localization using LipoGlo chemiluminescent microscopy. **a** Representative images of ApoB-LP localization patterns from analysis of 15 larvae per genotype from wild-type (WT), $mtp^{-/-}$, and $apoC2^{-/-}$ genotypes (6 days post fertilization (dpf)). The white arrow and arrowhead mark the larval intestine and liver respectively. **b** LipoGlo signal colocalizes with the central nervous system marker $Tg(Xla. Tubb2-mApple-CAAX)$, quantification in Supplementary Fig. 5. **c** LipoGlo signal localized to subregions of the central nervous system (CNS). **d–f** Representative images and **g–i** quantification of ApoB-LP localization across developmental, genetic, pharmacological, and dietary manipulations. **g** Signal localization at each day of larval development in WT (degree of freedom (DF) = 5, $n = 15$, Welch's analysis of variance (ANOVA) $p < 0.0001$ for each region over time), $mtp^{-/-}$ (DF = 11, $n = 15$, two-way robust ANOVA $p < 0.001$ for all regions, Games–Howell $p < 0.001$), and $apoC2^{-/-}$ (DF = 11, $n = 15$, two-way robust ANOVA was not significant for any region) genetic backgrounds. Upward-facing arrowheads (green) indicate significant enrichment of that species at that time point compared to WT, and downward-facing arrowheads (magenta) indicate depletion using the Games–Howell test with a threshold of $p < 0.05$. **h** Signal localization from 3 to 5 dpf in larvae treated with 10 µM lomitapide or vehicle control (DF = 11, $n = 15$, two-way robust ANOVA $p < 0.001$ for head and viscera, Games–Howell $p < 0.0001$). **i** Subclass abundance from 10 to 15 dpf in larvae subjected to a fasting and re-feeding paradigm. The first bracket delineates changes relative to time 0 (the onset of the fasting period), and the second bracket delineates changes relative to time point 48 (the onset of the re-feeding period) (DF = 10, $n = 15$, Welch's ANOVA $p < 0.0001$ for each region, Games–Howell $p < 0.005$). Supplementary Figure 4 displays standard deviations for **g–i**. Results represent pooled data from three independent experiments, "$n$" denotes number of samples per data point. Scale bars = 500 µm. Source data are provided as a Source Data file

cholesterol content. However, this approach has several important shortcomings in that (i) it is unable to detect lipoproteins outside of the bloodstream, (ii) it provides limited information on the size or abundance of atherogenic lipoproteins, and (iii) it is not conducive to high-throughput screening. LipoGlo enables direct assessment of lipoprotein abundance, size, and localization in high-throughput, thus circumventing several of the long-standing barriers to the study of atherogenic lipoproteins.

We elected to develop LipoGlo in the larval zebrafish model system because it is well suited for both high-throughput screening and in vivo imaging. Additionally, as it has not previously been possible to characterize atherogenic lipoproteins from individual larval zebrafish, the introduction of the LipoGlo reporter and characterization of the lipoprotein profile throughout development represent valuable resources for the zebrafish community. We go on to show that genetic mutations ($apoc2^{-/-}$

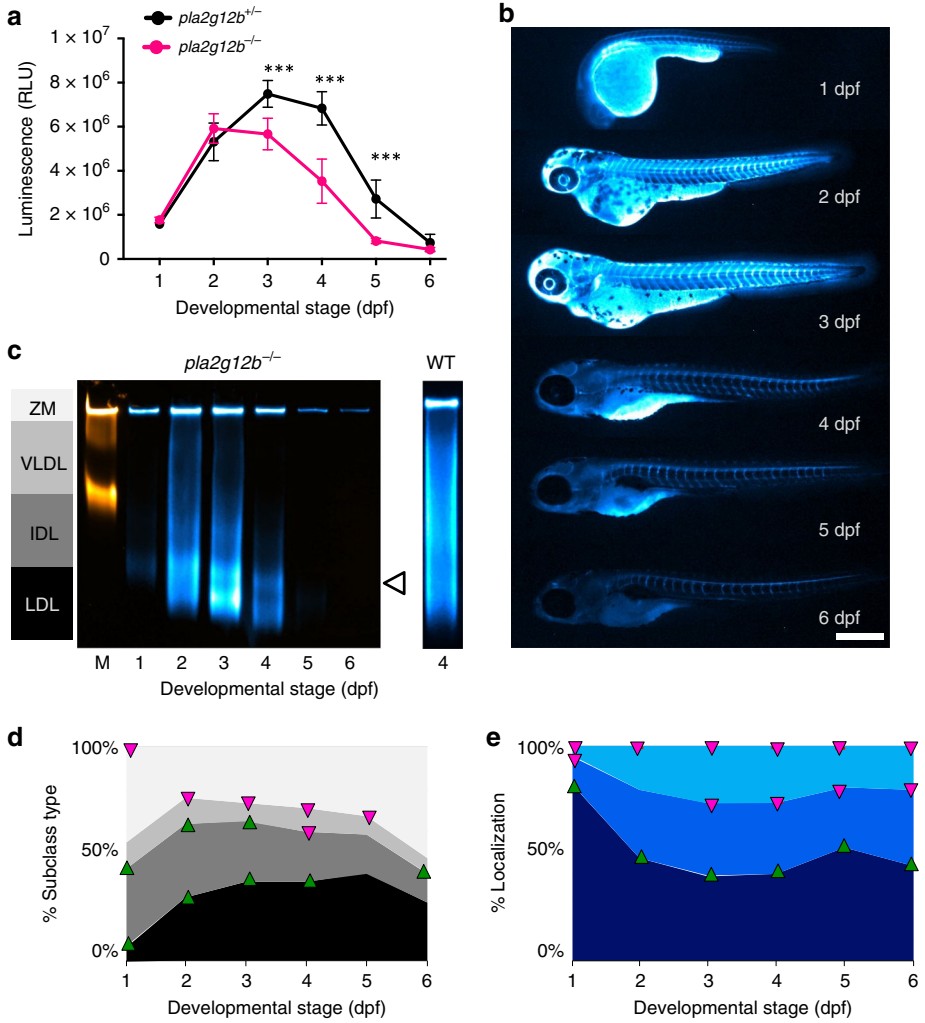

**Fig. 6** LipoGlo reveals profound alterations in the apolipoprotein B-containing lipoprotein (ApoB-LP) profile in *pla2g12b*−/− mutant larvae. **a** Comparison of LipoGlo signal between *pla2g12b*−/− and *pla2g12b*+/− siblings during larval development (1–6 days post fertilization (dpf)) (degrees of freedom (DF) = 11, $n \approx 11$, two-way robust analysis of variance (ANOVA) $p < 0.0001$ for genotype and stage, Games–Howell $p < 0.0001$). **b** Representative images ($n = 15$) of ApoB-LP localization collected by LipoGlo chemiluminescent imaging throughout development (1–6 dpf), and **e** quantification of percent localization into previously described subregions (DF = 11, $n = 15$, two-way robust ANOVA $p < 0.001$ for all regions, Games–Howell $p < 0.0001$). **c** Representative gel ($n = 4$) showing production of abnormally small lipoproteins (white arrowhead) and **d** quantification of LipoGlo emission pattern from Native-polyacrylamide gel electrophoresis (Native-PAGE) samples in *pla2g12b*−/− larvae (1–6 dpf). Upward-facing arrowheads (green) indicate significant enrichment of that species at that time point compared to WT, and downward-facing arrowheads (magenta) indicate depletion using the Games–Howell test with a threshold of $p < 0.05$ (DF = 11, $n = 9$, two-way robust ANOVA $p < 0.001$ for all species, Games–Howell $p < 0.01$). Results represent pooled data from three independent experiments, "*n*" denotes number of samples per data point. *** denotes p < 0.0001. Scale bars = 500 μm. Source data are provided as a Source Data file

and *mtp*−/−), pharmaceuticals (Lomitapide), and dietary manipulations (fasting and high-fat feeding) all affect the larval zebrafish lipoprotein profile just as they do in humans, suggesting highly conserved lipoprotein processing pathways between these two species.

As LipoGlo uses a protein tag (NanoLuc) to track atherogenic lipoproteins, it was essential to validate that the reporter remains attached to ApoB in vivo and does not disrupt lipoprotein homeostasis. Western blotting and DGUC were used to show that NanoLuc remains covalently attached to ApoB, and that NanoLuc signal is detectable in density fractions corresponding to atherogenic lipoproteins. Further, lipoprotein profiles of adult zebrafish (labeled with DiI) are indistinguishable between WT and LipoGlo animals. These findings establish a precedent for the use of ApoB-fusion proteins as a sensitive and specific approach to monitor atherogenic

lipoproteins, which will likely be generalizable to other model systems.

LipoGlo assays show excellent concordance with traditional methods, as evidenced by the tight correlation between particle size estimates measured by LipoGlo electrophoresis and both DGUC and negative-staining electron microscopy. However, one limitation of using crude extracts for LipoGlo electrophoresis is that several species above a certain size threshold cluster together as the ZM fraction. Further investigation is required to distinguish the specific species within the ZM fraction, but it is expected to include chylomicrons and chylomicron remnants[43], aggregated lipoproteins[44], and intracellular ApoB complexed with components of the secretory pathway (such as the ER, golgi, and other secretory vesicles)[45]. In line with these expectations, the ZM band is highly enriched in response to a high-fat meal, which likely reflects an increase in both nascent intracellular

lipoproteins and chylomicrons. Our study of lipoproteins in larval zebrafish using LipoGlo is not only highly concordant with previous work using traditional techniques in mammals but also led to several unexpected observations.

Mutations in the *mtp* gene result in a severe reduction or complete lack of ApoB-LPs, a disease called abetalipoproteinemia. The *mtp*[stl] allele studied here contains a single missense mutation in a highly conserved residue (L475P)[34]. As this is thought to result in production of a non-functional protein, we expected *mtp*[stl] homozygous mutants to exhibit a complete lack of ApoB-LPs. However, mutants are unequivocally able to produce and secrete ApoB-LPs (although they are smaller and less abundant) early in development. These observations suggest that the *mtp*[stl] allele is either a strong hypomorph or that ApoB-LPs can be produced without the activity of Mtp. LipoGlo will therefore serve as a useful tool to investigate additional alleles of *mtp*, such as a genomic deletion, to determine the precise role of Mtp in lipoprotein biogenesis. A deeper understanding of *mtp* alleles would be useful in predicting the outcome and potentially treating different abetalipoproteinemia mutations in humans. It was also unexpected to note that *mtp*[−/−] mutants produce a distinct bimodal peak of small (LDL-like) ApoB-LPs (Fig. 3a–c). This pattern warrants further investigation, but may indicate that these alleles directly produce small lipoproteins from the YSL, which are subsequently lipolyzed to produce a second peak. This observation warrants further investigation of the role of Mtp in regulating the size of nascent ApoB-LPs.

The distribution of atherogenic lipoproteins throughout an intact organism has not been comprehensively characterized, but we expected localization patterns to correspond primarily to lipoprotein-producing tissues (such as the liver, intestine, and yolk-syncytial layer) and the circulatory system. While we did indeed observe these expected patterns, we also observed a surprisingly high enrichment of LipoGlo signal associated with the central nervous system and the MS. The localization pattern observed in the central nervous system is strikingly similar to that of fluorescein (a fluorescent dye) following injection into the larval zebrafish ventricle[46], indicating that it may be present in the cerebrospinal fluid (CSF). While ApoB is nearly undetectable in the CSF of adults[47], high levels of ApoB have been detected in embryonic CSF, which play a role in body axis formation[48]. In addition to the CNS, lipoproteins also appear to be enriched in a chevron pattern outlining the somites in the trunk region. We suspect that this pattern may correspond to the MS[49], which include tendinous structures connecting the body segments and their associated lymphatic vessels[50]. Lipoproteins have previously been shown to accumulate in tendons in cases of severe hyperlipidemia[51], but LipoGlo imaging suggest that a large fraction of ApoB-LPs localize to tendinous structures in a normal physiological state as well. Together, these observations suggest that ApoB-LPs may play important physiological roles outside the circulatory system. LipoGlo is an essential tool for broadening the scope of atherogenic lipoprotein biology beyond the current focus on circulating particles.

Phospholipase A2 group XII B (*pla2g12b*) encodes a catalytically inactive member of the phospholipase gene family. Although the protein lacks catalytic activity and has no other known function, its high level of evolutionary conservation suggests it may have evolved a new function. Previous studies in mice have shown that disruption of *pla2g12b* results in decreased secretion of hepatic triglyceride and ApoB[52], as well as reduced levels of high-density cholesterol[53], indicating that this gene may play a role in lipoprotein secretion. In LipoGlo assays, *pla2g12b*[−/−] mutant larvae exhibited significantly lower levels of ApoB at multiple stages, and show enrichment of visceral ApoB-NanoLuc levels, both of which are consistent with previously reported

defects in lipoprotein secretion[52]. However, evaluation of the lipoprotein size distribution in *pla2g12b*[−/−] mutants revealed bias towards production of small LDL-like particles, implicating Pla2g12b as a previously unappreciated regulator of lipoprotein particle size.

Here we demonstrate that tagged ApoB enables highly sensitive and specific monitoring of ApoB-LPs without disrupting lipoprotein homeostasis. Characterization of the lipoprotein profile of larval zebrafish using LipoGlo revealed human-like responses to genetic, dietary and pharmacological stimuli, establishing larval zebrafish as a tractable vertebrate system to study lipoprotein homeostasis. Studying lipoproteins in zebrafish also created opportunities for unbiased screening and whole-organism imaging, which led to the identification of Pla2g12b as a regulator of lipoprotein particle size and the detection of significant extravascular lipoproteins. LipoGlo thus represents a powerful approach to expand our understanding of atherogenic lipoproteins and accelerate the discovery of additional genes and small-molecule probes that modulate ApoB homeostasis.

## Methods

**Zebrafish husbandry and maintenance**. Adult zebrafish were maintained on a 14 h light–10 h dark cycle and fed once daily with ~3.5% body weight of Gemma Micro 500 (Skretting USA). All genotypes were bred into the WT AB background. All assays were performed on larvae heterozygous for the ApoB-Nanoluc reporter, unless otherwise noted. To monitor the WT lipoprotein profile throughout larval development, pairwise crosses were set up between WT AB adults and adults homozygous for the ApoB-NanoLuc reporter (*apoB.1*[NLuc/NLuc]). To characterize the lipoprotein profile of *mtp* mutant larvae[34], pairwise crosses were set up between *mtp*[stl/+] and *mtp*[stl/+]; *apoBb.1*[NLuc/NLuc] adults. To characterize the lipoprotein profile of *apoC2* mutant larvae[17], pairwise crosses were set up between *apoC2*[sd38/sd38] and *apoC2*[sd38/+]; *apoBb.1*[NLuc/+] adults and larvae positive for the NanoLuc reporter were selected for analysis. To characterize the lipoprotein profile of *pla2g12b* mutant larvae[42], pairwise crosses were set up between *pla2g12b*[sa659/sa659] and *pla2g12b*[sa659/+]; *apoBb.1*[NLuc/+] adults. To evaluate association between the ApoB-LPs and the central nervous system, adults homozygous for the ApoB-NanoLuc reporter (*apoBb.1*[NLuc/NLuc]) were crossed to adults heterozygous for the central nervous system marker *Tg(Xla.Tubb2:mapple-CAAX)*, and embryos were screened for mApple prior to fixation and mounting (unpublished reagent provided by the Halpern Lab, c583). As zebrafish sex cannot be determined during the larval stages, gender can be excluded as a variable. The key resources table (Supplementary Table 2) details the sources for all transgenic lines. All procedures comply with all relevant ethical regulations and were approved by the Carnegie Institution Animal Care and Use Committee (Protocol #139).

**Genome editing**. Genome integration was achieved by co-injection of 500 pg of TALEN mRNA and 30 pg of donor plasmid into one-cell stage embryos (Supplementary Fig. 2a). Two pairs of TALENs were designed and cloned that target a *Bsr*I restriction site just upstream of the endogenous stop codon of *apoBb.1* using the Mojo Hand design tool[54] and FusX assembly system[55]. TALENs were in vitro transcribed using the T3 Message Machine Kit (Thermo Fisher Scientific, AM1348) and injected into one-cell stage zebrafish embryos. Cutting efficiency was quantified by monitoring the loss of *Bsr*I digestion as a result of TALEN nuclease activity, and found to be significantly higher in TALEN pair 2, so this pair was used for genome integration efforts (Supplementary Fig. 2b). A donor plasmid was cloned using three-fragment MultiSite gateway assembly (Invitrogen, 12537-023) with a 5′ entry element of ~500 bp of the genomic sequence upstream of the *apoBb.1* stop codon, a middle-entry element consisting of in-frame NanoLuc coding sequence, and a 3′ element of ~700 bp of genomic sequence downstream of the *apoBb.1* stop codon[56]. Injected embryos were raised to adulthood and progeny were screened for NanoLuc activity and in-frame fusion of the NanoLuc reporter at the target locus (Supplementary Fig. 2c). The key resources table (Supplementary Table 2) details the sources for and Addgene deposition numbers for all plasmids used in this study.

**Preparation and storage of larval homogenate**. Individual larvae are homogenized in a standard volume of ApoB-LP stabilization buffer (100 µL). The ApoB-LP stabilization buffer (recipe below) contains cOmplete Mini, EDTA-free Protease Inhibitor Cocktail (Millipore-Sigma, 11836170001), pH buffer and calcium chelator (EGTA, pH 8), and cryoprotectant[57] (sucrose) to preserve sample integrity during homogenization (Supplementary Fig. 6). The buffer is made as a 2× stock, and larvae are anesthetized in tricaine and placed into tubes in a 50 µL volume and an equal volume of chilled 2× buffer is then added just prior to homogenization. Low-throughput homogenization can be achieved in 1.5 mL centrifuge tubes with disposable pellet pestles (Thermo Fisher Scientific, 12-141-363). For high-

throughput sample processing, larvae and ApoB-LP stabilization buffer are dispensed into individual wells of a 96-well non-skirted PCR-plate (USA Scientific, #1402-9589), sealed with microSeal "B" plate sealing film (Bio-Rad, msb1001), and homogenized in a microplate-horn sonicator (Qsonica, Q700 sonicator with 431MPX microplate-horn assembly). For sonication, the plate was placed in the microplate-horn filled with 17 mm of chilled reverse osmosis (RO) water and processed at 100% power for a total of 30 s, delivered as 2-s pulses interspersed with 1-s pauses. Homogenate was stored on ice for immediate use, or frozen at −20 °C and thawed on ice for later use.

For routine preparation of zebrafish homogenate, a 2× solution of ApoB-LP stabilization buffer was prepared by mixing 1 cOmplete Mini Protease Inhibitor Tablet, 1 g of sucrose, and 400 μL of 0.5 M EGTA (pH 8) and adjusting the volume to 5 mL with RO water. Plate-based measurement of NanoLuc activity were performed using 2× NanoLuc buffer, prepared by mixing 1 mL of Nano-Glo buffer, 3 mL phosphate-buffered saline (PBS), and 20 μL NanoLuc Substrate (furimazine solution).

**Quantification of ApoB-NanoLuc levels using a plate reader**. To quantify ApoB-NanoLuc levels, homogenate (40 μL) was mixed with an equal volume of 2x NanoLuc buffer (recipe above) in a 96-well opaque white OptiPlate (Perkin-Elmer, 6005290). Black plates can be used as an alternative that will significantly lower absolute signal intensity, and also reduce light contamination into adjacent wells. The plate was read within 2 min of buffer addition using a SpectraMax M5 plate reader (Molecular Devices) set to top-read chemiluminescent detection with a 500 ms integration time. This plate-based assay has a wide linear range and long half-life (Supplementary Fig. 7a–c). However, degree of pigmentation has a significant effect on signal intensity, so this variable should be accounted for with a standard curve or pigment-matched controls should be used as a baseline for comparison (Supplementary Fig. 7d).

**Quantification of lipoprotein size distribution with LipoGlo electrophoresis**. To quantify the electrophoretic mobility of ApoB-LPs, 3% Native-polyacrylamide gels were cast in Bio-Rad mini-protean casting rigs using 1 mm spacer plates and 10-well combs (recipe below). Gels were allowed to polymerize overnight at 4 °C and used within 24 h of casting. Each gel included a migration standard comprised of DiI-labeled human LDL (L3482, Thermo Fisher Scientific) that was diluted in a cryoprotectant and stored in frozen aliquots (recipe below). Gels were assembled into mini-protean electrophoresis rigs at 4 °C, filled with pre-chilled 1× TBE and pre-run at 50 V for 30 min to equilibrate the gel prior to sample addition. Twelve microliters of homogenate was then combined with 3 μL of 5× load dye (recipe below), and 12.5 μL of the resulting solution was loaded per well (which corresponds to 10% of the larval homogenate per lane). Gels were then run at 50 V for 30 min, followed by 125 V for 1 h.

Gels were imaged within 1 h of completion of the run. To image each gel, the thin glass short plate was carefully separated from the front of the gel with a gel releaser wedge. With the gel resting on the thick spacer plate, 1 mL of TBE supplemented with 2 μL of Nano-Glo substrate was gently pipetted onto the gel surface. The gel imaging solution was spread evenly across the gel surface with a thin plastic film cut to the size of the spacer plate (Staples, Sliding bar report covers). After a 5-min equilibration, the gel was placed into an Odyssey Fc (LI-COR Biosciences) gel imaging system and imaged in the chemiluminescence channel for 2 min (NanoLuc detection) and then the 600 channel for 30 s (DiI LDL standard detection). Effort should be made to ensure a consistent equilibration time between substrate addition and gel imaging. While this assay is relatively robust to small variations in incubation time (Supplementary Fig. 7g–h), equilibration times below 5 min or >20 min are not recommended. Raw images were exported as zip files for further analysis.

The provided gel quantification template (Supplementary Software 1) can be used to bin the complex lipoprotein size distribution into biologically relevant groups for analysis, and detailed instructions are provided within the supplemental file. In short, each lane was converted to a plot profile in ImageJ, and divided into LDL, IDL, VLDL, and ZM bins based on migration relative to the DiI standard, and pixel intensity was summed within each bin for analysis.

The most likely source of artifacts in the LipoGlo-electrophoresis protocol are from stretching or distortion of the fragile 3% polyacrylamide gel while removing the short plate from the gel. To circumvent this issue, the short plates were coated on both sides with Rain-X original glass water repellent (Rain-X, 3.5 oz bottle). This hydrophobic coating greatly facilitates removal of the short plate while leaving the undistorted gel resting on the spacer plate. This coating is semi-permanent, so it is recommended that a set of coated short plates be dedicated for this purpose and reapplied with coating as needed. This hydrophobic coating also reduces friction between the short plate and the spacer plate, so it is important that the plates are aligned properly in the casting frames and placed very gently in the casting stands. Too much pressure from the casting stand can cause the plates to slide out of alignment and lead to leaking during casting.

For LipoGlo electrophoresis of ApoB-LPs from larval homogenate, 3% Native polyacrylamide gels were cast by mixing 22.9 mL of RO water, 6.4 mL of 5× TBE, 2.4 mL of 19:1 polyacrylamide:bis solution (40%), and de-gassing the solution under vacuum for 30 min. This volume of solution (~32 mL) is sufficient to cast four mini-protean gels. Once the casting stands are assembled, 250 μL of 10% APS

(ammonium persulfate) and 20 μL of TEMED (tetramethylethylenediamine) were added to the solution and mixed gently by inversion, and then immediately pipetted into casting plates.

DiI LDL was used for normalization of electrophoretic mobility in LUs. Two hundred microliters of DiI LDL (L3482, Thermo Fisher Scientific) was diluted into 4 mL of 1× TBE, and then 0.48 g of sucrose was dissolved into this solution to serve as a cryoprotectant. The final volume was adjusted to 4.8 mL with 1× TBE and the solution was divided into 50 μL aliquots and stored at −80 °C. Loading dye (5×) for loading larval homogenate and DiI standard into LipoGlo-electrophoresis gels was prepared by dissolving 4 g sucrose and 25 mg bromophenol blue in a final volume of 10 mL TBE. In-gel chemiluminescent imaging of NanoLuc was performed by adding 1 mL TBE containing 2 μL furimazine substrate solution to the gel surface.

**Larvae fixation and imaging**. To determine the whole-organism localization of ApoB-LPs, intact larvae are anesthetized and fixed in 4% PFA (diluted in PBS) for 3 h at room temperature. Following fixation, larvae are rinsed three times for 15 min each in PBS-Tween (PBS containing 0.1% Tween-20 detergent) and imaged within 12 h of fixation. Agarose for mounting is prepared by melting 0.1g of low-melt point agarose (BP160-100, Thermo Fisher Scientific) in 10 mL of 1× TBE. Aliquots are maintained in the liquid state at 42 °C in a heat block. Just prior to mounting, agarose aliquots were supplemented with 1% Nano-Glo substrate (furimazine). Fixed larvae are arrayed in droplets on a Petri dish lid, and the excess liquid is removed and quickly replaced with a 50 μL droplet of low-melt agarose containing Nano-Glo substrate (1%). The sample is then oriented properly with a flexible poker until the agarose solidifies sufficiently to hold the sample in place. This process was repeated for up to 15 larvae in parallel prior to imaging.

While the absolute intensity of NanoLuc signal will decay gradually over time (Supplementary Fig. 7e–f), the distribution of NanoLuc signal is relatively robust to slight variation in incubation time. Nonetheless, it is recommended that larvae be mounted and imaged with as consistent of an incubation period as possible (ideally <30 min).

To image the ApoB-LP localization, a Zeiss Axiozoom V16 microscope V16 equipped with a Zeiss AxioCam MRm was set to ×30 magnification, 2 × 2 binning and 2× gain (to increase sensitivity), and programmed to collect a single brightfield exposure (2.4 ms, 10% light intensity), followed by two chemiluminescent imaging exposures (10 and 30 s, respectively) with no illumination to collect the NanoLuc signal. Images were quantified in ImageJ by using the brightfield exposure to draw ROI (viscera, trunk, and head) and calculating the NanoLuc intensity within each of those ROIs for 30 s chemiluminescent exposure, unless saturated pixels were detected in which case the 10 s exposure was used.

Essentially all background signals in this imaging paradigm comes from two sources: electrical noise from the camera and light contamination from the environment. Camera noise can be attenuated by using an actively cooled camera and by enabling a blank-subtraction setting to eliminate hot pixels. To reduce contaminating light from the environment, we recommend collecting images in a dark room and shrouding the stage and/or microscope to prevent light from reaching the imaging path. Additionally, we have found that the Zeiss Axiozoom V16 contains infrared emitters and detectors within the imaging path, which result in very high background when long exposures are used. To overcome this issue, we placed a Zeiss BG40 IR blocking filter in front of the camera which effectively filtered the contaminating infrared light.

For imaging of ApoB-LP distribution in intact larvae, mounting and imaging solution was prepared by dissolving 0.1 g low-melt agarose into 10 mL of 1× TBE and heating in the microwave for 5–15 s. One milliliter of aliquots of the liquid gel were then distributed into 1.5 mL centrifuge tubes and maintained as liquid in a 42 °C heat block for up to 1 week. Ten microliters of furimazine was then added to 1 mL liquid agarose just prior to mounting. Larvae are mounted in a volume of 50 μL, so a single aliquot is sufficient for ~20 larvae.

**Fluorescent labeling and imaging of APOB-LPs in vivo**. For injection of human DiI-labeled LDL into zebrafish larvae, larvae were first anesthetized and mounted laterally in a petri dish in a 50 μL droplet of 1% low-melt agarose prepared in zebrafish embryo medium. After the agarose had solidified, forceps were used to remove a small portion of agarose just dorsal to the anterior portion of the yolk, providing an access window for injection. The Petri dish was then filled with embryo medium to ensure that larvae did not dry out during injection and imaging. Human DiI-labeled LDL (L3482, Thermo Fisher Scientific) was then loaded into a microinjection needle and calibrated to an injection volume of 4 nL. Larvae were then injected into the common cardinal vein through the agarose-free access window, or injected directly into the yolk. Larvae were imaged within 1 h of injection, and then carefully liberated from the agarose using fine forceps.

For injection of DiI into the larval yolk, DiI (D282, Thermo Fisher Scientific) was resuspended to 30 mg/mL in dimethyl sulfoxide. This solution was then loaded into an injection needle and calibrated to 4 nL injection volume. One day post fertilization was selected as an ideal time point for injection because at this stage the yolk is fully segregated from the embryo, yet the embryos are still within their chorions, which eliminates the necessity for mounting in agarose prior to injection. Larvae were then imaged on an SMZ25 microscope equipped with a Cy3 filter cube at ×30 or ×100 magnification. Larvae were imaged after being mounted in either 1% low-melt agarose (in the case of 2 dpf injections) or were anesthetized

and imaged in 3% methylcellulose. Image brightness was adjusted arbitrarily in the display to ensure visibility of relevant structures.

**DiI labeling of adult zebrafish plasma.** Adult zebrafish were anesthetized and transferred to a dissection stage covered with a kimwipe moistened with anesthetic diluted in system water. Tails were then resected at the anterior of the anal fin to expose the CA. An EDTA-coated capillary (Thermo Fisher Scientific, 22-757-123) fitted on the end of a p20 pipette tip was used to slowly extract whole blood form the artery, and the fish was immediately euthanized in an ice bath following extraction. Blood was transferred to a 1.5 mL microcentrifuge tube and spun at $6000 \times g$ for 5 min to pellet blood cells, and the resulting plasma was transferred to a new tube. The plasma was then diluted ten-fold in a solution of ApoB-LP stabilization buffer containing 30 mg/mL DiI. This mixture was incubated at 37 °C for 2 h, and stored at 10 °C overnight or used immediately for electrophoresis. Five microliters of the resulting stained lipoproteins were mixed with 45 μL of ApoB-LP buffer containing loading dye and 12.5 μL of the mixture was loaded onto a 3% Native-PAGE. Native-PAGE and gel imaging were performed as described above for LipoGlo electrophoresis.

**Density gradient ultracentrifugation.** A DGUC protocol was developed by adapting previously published protocols using a 3-layer iodixanol gradient to function with smaller volumes of input sample[39]. Individual larvae were sonicated in 100 μL of sucrose-free ApoB-LP buffer (recipe below) to avoid disruption of the density gradient with sucrose. Fifteen larvae were pooled per experiment into a single 1.5 mL centrifuge tube and centrifuged for 5 min at $6000 \times g$ to remove large cellular debris. One milliliter of the resulting supernatant was transferred to a separate tube containing 500 μL of OptiPrep Density Gradient Medium (D1556, Sigma-Aldrich) to yield a 20% iodixanol solution. A 9% iodixanol solution was prepared by adding 1.5 mL of OptiPrep to a 15 mL conical tube containing 8.5 mL HEPES-buffered saline (HBS, recipe below), and a 12% solution was prepared by mixing 2 mL OptiPrep with 8 mL HBS. A 4.9 mL Optiseal tube (formerly poly-allomer, 362185, Beckman-Coulter) was then loaded with 1.5 mL of 9% iodixanol/ HBS solution. This solution was carefully underlayered with 1.5 mL of the 12% iodixanol solution using a p1000 pipette fit with both the appropriate p1000 tip as well as a tapered gel loading tip, which functioned as a disposable plastic cannula (USA Scientific, 1252-0610). Finally, these solutions were underlayered with 1.5 mL of the 20% iodixanol solution containing the zebrafish homogenate. The tube was then topped up with HBS (~500 μL) so that no air remained and then was sealed with a cap. Balanced tubes were then loaded into a VTi65.2 rotor and centrifuged at $316{,}000 \times g$ in a pre-chilled Beckman Optima XL 80K Ultracentrifuge set to 4 °C with maximum acceleration and deceleration rates.

Following ultracentrifugation, density fractions were collected by carefully piercing the bottom of the tube with a thumbtack, and then drip eluting the samples into 10 separate fractions of approximately 500 μL each. The refractive index of each fraction was determined using a Bausch and Lomb refractometer, and used to calculate solution density using the formula density = 3.3508 × (refractive index) − 3.4675. Fractions were stored on ice or at 10 °C, and used within 24 h for a plate-based NanoLuc assay, LipoGlo electrophoresis, and negative-staining electron microscopy. Note that the high protein and iodixanol content of fraction 1 (highest density) introduces artifacts in the Native gel and was therefore excluded, which allowed lane 1 to be dedicated to the DiI LDL standard.

A 2× solution of sucrose-free ApoB-LP stabilization buffer was prepared for processing zebrafish homogenate for ultracentrifugation by mixing 1 cOmplete Mini Protease Inhibitor Tablet and 400 μL of 0.5 M EGTA (pH 8), and adjusting the final volume to 5 mL with RO water. HBS was used for dilution of OptiPrep Density mMdium, and was prepared by mixing 0.85 g NaCl, 10 mL 1 M HEPES buffer (pH 7.4), and 90 mL RO water.

**Negative-staining electron microscopy.** Fractions 4, 7, and 10 from the DGUC experiments outlined above were subjected to negative-staining electron microscopy[40]. Three hundred-mesh copper grids coated with 10 nm formvar and 1 nm carbon (Electron Microscopy Sciences, FCF300-Cu) were ionized using the glow discharge filament in a Denton Vacuum dv-502 evaporator at 75 mTorr for 30 s. Anti-capillary forceps were then used to hold the grids in a humidified chamber, and 3 μL of the sample was carefully placed on the surface of the grid and incubated at room temperature for 10 min to allow the lipoproteins to adhere to the grid. The grid was then rinsed in five droplets of RO water and then finally two droplets of 2% uranyl acetate, and finally touched lightly to a piece of filter paper to remove excess stain. Grids were imaged at ×26,000 magnification on a Tecnai 12 transmission electron microscope.

**Western blotting.** Protein extraction was performed on 10 pooled 3 dpf larvae per sample. Larvae were transferred to a 1.5 mL microcentrifuge tube, excess liquid was removed, and then 100 μL of RIPA buffer containing 3× protease inhibitor cocktail was added. Larvae were immediately homogenized using a pellet pestle, and incubated at 4 °C for 15 min with shaking. Samples were then centrifuged at $12{,}000 \times g$ for 5 min and the supernatant was mixed with an equal volume of 2× Laemmli buffer (Bio-Rad, 1610737) and heated to 95 °C for 5 min in a thermal cycler. DiI-LDL (L3482, Thermo Fisher Scientific) was diluted 100-fold in RIPA

buffer and extracted as above to be used as an indicator of the migration pattern of APOB, and Halo-Tagged NanoLuc protein (Promega, CS188401) was diluted 10,000-fold in RIPA buffer and used as an indicator of the migration of free NanoLuc protein. Precision Plus Protein All Blue Prestained Protein Standards (Bio-Rad, 1610373) was used as a molecular weight marker.

Twenty-five microliters of the resulting sample was loaded onto a precast 4–20% gradient gel (Bio-Rad, 4561093) and separated at 70 V for 30 min and 90 V for 60 min. Proteins were then transferred to a PVDF membrane with the Trans-blot Turbo Transfer System (Bio-Rad, 1704150) using a custom transfer program optimized to ensure transfer of high-molecular weight proteins (1.3 A constant for 15 min). The blot was blocked in 5% milk for 1 h, and then probed simultaneously with primary antibodies binding NanoLuc (R&D Systems, MAB10026-100, 1:200 dilution) and human APOB (Meridian Life Sciences, K45253G, 1:200 dilution) for 4 h at room temperature in 2.5% milk. The blot was then rinsed three times for 5 min each in TBST, and probed with fluorescent secondary antibodies (LI-COR Biosciences, IRDye 800CW Donkey Anti-Goat IgG, 925-32214, and IRDye 680RD Donkey Anti-Mouse IgG, 925-68072, 1:5000 dilution) for 1 h at room temperature. The blot was then rinsed as above and imaged in the 700 and 800 nm channels for 2 min each using the Odyssey Fc (LI-COR Biosciences).

A 15 mL solution of RIPA buffer was prepared for homogenization of zebrafish larvae for western blotting, comprised of 1.5 mL of 10% NP-40, 1.5 mL of sodium deoxycholate, 1.5 mL of 10% SDS, 2.25 mL of 1 M NaCl, 1.5 mL of 0.1 M sodium phosphate, 0.3 mL of 0.1 M EDTA, 1.8 mL PI cocktail (1 cOmplete Mini Protease Inhibitor Tablet dissolved per mL water), 75 μL of 200 mM PMSF, 150 μL of 100 mM sodium orthovanadate, and 5.8 mL RO water.

**DNA extraction and genotyping.** Sonication of zebrafish larvae is a convenient method for highly parallelized homogenization, as a full plate (96 samples) can be processed simultaneously. However, this process shears DNA into significantly smaller fragments, meaning longer amplicons will amplify less efficiently or not at all. To circumvent this issue, genotyping protocols for this study were designed to use small amplicons (<350 bp). If intact DNA is needed for downstream applications, the pellet-pestle method can be used interchangeably with sonication.

DNA extraction of larval homogenate can be achieved with a modified version of the HotShot DNA extraction protocol[58]. Ten microliters of homogenate is transferred to a PCR tube/plate containing 10 μL of 100 mM NaOH, and then heated at 95 °C for 20 min. The solution was then neutralized with an equal volume (20 μL) of 100 mM Tris, pH 8, and either stored frozen (−20 °C) or used immediately as a template for genotyping PCR (2 μL per reaction).

Genotyping was carried out using gene-specific primers (Supplementary Table 1). The *apoBb.1-NanoLuc* locus was genotyped using three primers with final concentrations as follows: 1 μM primer 9, 0.2 μM primer 10, and 0.8 μM primer 11. This ratio provides similar band intensity for the 113 bp product indicating the presence of the WT allele, and the 161 bp product indicating NanoLuc fusion allele ($T_a = 57$ °C, extension time 20 s) in heterozygotes (only one band will amplify in homozygotes). The *mtp* genotyping locus was amplified using primers 12 and 13 (0.5 μM each, $T_a = 60$ °C, extension time 30 s), and digested with 3 U of *Ava*II restriction enzyme, which cuts the mutant (*stl*I) allele. WT zebrafish should have a single 157 bp band, homozygous mutants should have a shorter 129 bp band, and heterozygotes should have both bands present (note the 28 bp fragment is not usually detectable). The *apoC2* genotyping locus was amplified using primers 14 and 15 (0.5 μM each, $T_a = 57$ °C, extension time 30 s), and digested with 3 U of *Bts*αI restriction enzyme, which cuts the WT allele but not the sd38 mutant allele. WT zebrafish should have 102 and 45 bp bands, homozygous mutants should have a single 147 bp band, and heterozygotes should have all three bands present. The *pla2g12b* genotyping locus was amplified using primers 16 and 17 (0.5 μM each, $T_a = 57$ °C, extension time 30 s), and digested with 3 U of *Bts*αI restriction enzyme, which cuts the mutant (sa659) allele. WT zebrafish should have a single 150 bp band, homozygous mutants should have a shorter 111 bp band, and heterozygotes should have both bands present (note the 39 bp fragment is not usually detectable).

**Quantification and statistical analysis.** All datasets were initially subjected to Levene's test for homogeneity of variance. For datasets with a single factor and uniform variance, a one-way analysis of variance (ANOVA) was used to test for a main effect, and Tukey's honestly significant difference (HSD) was used for post hoc testing. If variance was not uniform (Levene's < 0.05), Welch's ANOVA with a post hoc Games–Howell test was used as these tests are robust to the assumption of unequal variance. For two-factor datasets, the robust two-factor ANOVA was used with a post hoc Games–Howell test. * denotes $p < 0.01$, ** denotes $p < 0.001$, and *** denotes $p < 0.0001$. For LipoGlo-electrophoresis experiments, statistical tests were run independently for each of the four groups of binned data (ZM, VLDL, IDL, and LDL). In this case, Bonferroni correction was used to adjust for multiple comparisons (corrected significant $p < 0.0125$). Bonferroni correction was also applied to the LipoGlo-microscopy experiments, which are binned into three groups, so a significant threshold was set at $p < 0.017$. All statistics were run using XLSTAT, with the exception of the robust two-factor ANOVA, which was executed in R using the pbad2way function in the WRS2 package (https://cran.r-project.org/web/packages/WRS2/index.html).

One of the strengths of a chemiluminescent reporter is that it has excellent signal to background ratios. In our hands, the signal-to-background ratio varies

significantly between assays, reaching approximately 300,000:1 in plate-based assays, 30:1 in-gel-based assays, and 13:1 in microscopy assays (Supplementary Fig. 7i). Blank subtraction was therefore not performed for the above analyses, as it was found not to have an impact on the results. This is likely due to the fact that background signal is negligible in the plate-based assays, and the remaining assays not only have very low levels of background signal, but are also quantified in terms of relative (rather than absolute) signal and thus less susceptible to skewing from the background signal.

**Reporting summary**. Further information on research design is available in the Nature Research Reporting Summary linked to this article.

## Data availability

The datasets generated during and/or analyzed during the current study are available from the corresponding author on reasonable request. The source data underlying Figs. 2a–e, 3a–h, 4a–e, 4g, 5a–i, 6a–e and Supplementary Figs. 2b–d, 3a, 3c–g, 5a–e, 6a–b, 7a–i, and 8 are provided as a Source Data file.

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

## Acknowledgements

We thank Promega Corp. for providing the NanoLuc plasmid, as well as sample reagents and technical advice that were essential to assay development, as well as serving as a co-sponsor for a local lipid research conference where this work was presented. We would like to thank Michael Sepanski for collecting the electron micrographs, Dr. Michael McCaffery for technical support with negative-staining electron microscopy, Dr. Marnie Halpern for providing the unpublished *Tg(Xla. Tubb2:mapple-CAAX)* pan-neuronal marker line and valuable advice on the manuscript, Dr. Yury Miller for providing the *apoC2* mutant line, and the Sanger Institute Zebrafish Mutation project for providing the *pla2g12b* mutant line (sa659). We would also like to thank Robert Waddail for technical advice in developing the gel quantification template (Supplementary Software 1). Support was also provided by the National Institutes of Health (R01DK093399 [S.A.F.] and R01DK116079 [S.A.F.]), National Heart, Lung, and Blood Institute (F31HL139338 [J.H.T]), and National Institute of General Medical Sciences (R01GM63904 [S.C.E] and [S.A.F] and P30DK084567 [S.C.E.]). This content is solely the responsibility of the authors and does not necessarily represent the official views of NIH. Additional support for this work was provided by the Carnegie Institution for Science endowment and the G. Harold and Leila Y. Mathers Charitable Foundation (S.A.F).

## Author contributions

J.H.T. and S.A.F. conceived and designed the project, and met frequently to discuss results, plan experiments, and troubleshoot protocols. S.C.E. provided critical reagents and expertise to design and synthesize the TALENs used to create the LipoGlo fish line. J.H.T. executed the experiments, analyzed the results, and wrote the original draft of the paper. J.H.T., S.C.E. and S.A F. revised, edited, and approved the final submitted version of the manuscript.

## Additional information

**Competing interests:** The authors declare no competing interests.

