## [Peer Review File · Nature Communications]

Reviewers' comments:

Reviewer #1 (Remarks to the Author):

The authors develop a new method to track and image distribution of ApoB-containing lipoproteins based on a genetic fusion with the luciferase enzyme, NanoLuc. With this genetic reporter, they present assays for analyzing total amount of ApoB lipoproteins, particle types based on electrophoresis, and distribution in vivo based on imaging fixed zebrafish. They establish the approach can be applied to testing mutants and compounds. While NanoLuc has been fused to other proteins and used for imaging studies in vitro and in vivo, the described application is novel and applicable to relevant biologic questions. The presented in vivo images are convincing. Statistics and methods are appropriate to allow investigators to reproduce the imaging methods. Specific comments are listed below:

1. In the discussion, the investigators state that NanoLuc predominantly remains attached to ApoB. This statement needs quantitative support and documentation as integrity of the fusion protein in all assays and in vivo is critical to interpretation of data.
2. The authors should clarify the magnitude of background signal in assays relative to signal produced by NanoLuc-ApoB. Did the authors subtract background values from presented data in graphs?
3. The authors should comment on stability of the NanoLuc signal over time in various assays as this is important for reliable quantification.

Reviewer #2 (Remarks to the Author):

This manuscript from Thierer et al., described the generation of a new knockin zebrafish in which the apoB protein is tagged with luciferase (NanoLuc). The authors showed that this novel genetically modified animal can be used to track some of metabolic features of apo-B containing lipoproteins in a spatiotemporal manner, using mtp or apoc2 loss-of-function animals or using a MTP inhibitor. Indeed, few, if there is any previous technique can track the size of apoB lipoproteins in vivo using live animals. This platform can also be used to screen compounds interfering with lipoprotein metabolism in a large scale and be adopted to other organisms. Furthermore, leveraging this novel animal model, they defined the new function of pla2g12b as a critical regulator of apoB-containing lipoprotein size. In general, this is a very interesting paper demonstrates the powerful usage of

zebrafish animal model in combination with versatile gene targeting. The manuscript was well-written. While this is a well-conducted study, a few improvements may further strength their findings.

Major concerns:

1. As far as we know, most lipoproteins reside in the vasculature. The in vivo luciferase signal did not show enrichment in the blood vessels (Figure. 5a-d) but instead in the brain, spinal cord, liver and intestine. What would be the in vivo distribution of apoB-lipoproteins without the luciferase tag? The author may inject human fluorescently labeled human LDL and monitor their distribution.
2. What is the lipoprotein profile of the adult LipoGlo animal compared to unmodified AB animals, in the presence and absence of hyperlipidemia? Such as the LDL-C and HDL-C as measured in Ohare et al., *J Lipid Res.* 2014 Nov;55(11):2242-53
3. Figures 3 and 4 show differential apoB lipoprotein fractions using native gel electrophoresis and ultracentrifugation-based isolation. A verification of the presence of cholesterol using NBD cholesterol staining (or TopFLuo cholesterol-containing food feeding) and apoB using WB can further strength the conclusions.
4. The author identified the novel pla2g12b gene in modifying lipoprotein size. What is the current study on this gene? Any association of this gene with cardiovascular disease and lipoprotein metabolism? The author may test if indeed pla2g12b deficiency augments foam cell formation in apoC2 mutant or control animals.

Minor issue.

1. It is interesting that the animals were PFA fixed first and then the luciferase substrate was added. Is it because of high sensitivity of this NanoLuc?
2. What changes in the apoB containing lipoproteins when their size become smaller and render them more atherogenic? Any in vitro or in vivo evidence?

Reviewer #3 (Remarks to the Author):

This manuscript describes experiments to develop and validate the LipoGlo reporter for apoB containing lipoproteins in zebrafish. The authors have tagged the zebrafish homolog of apoB with a luciferase tag which allows the tracking of its lipoproteins around the transparent fish larvae. In addition, the larvae can be ground up and the lipoproteins analyzed by electrophoretic techniques to determine general distributions between CM, VLDL, IDL and LDL. The authors used the system to

identify potentially new sites of apoB-LP association within the larval body and, as a proof of concept, identified a gene that likely affects apoB-LP metabolism in fish. Overall, this is an interesting system. The resolution of the system, both in terms of the gel electrophoresis size analysis and the whole body localization studies, is perhaps a bit disappointing. I was hoping that it might be possible to see associations that are more related to atherosclerosis development like accumulation of apoB-LPs in the vasculature, for example. However, it clearly has potential uses in screening for genes and drug treatments that affect apoB-LP metabolism. However, there were some deficiencies in the characterization of the system and inconsistencies in the data that reduced enthusiasm for this work in its current state. My specific comments are outlined below.

- 1) The abbreviation ABCLs seems a bit clunky for apoB containing lipoproteins. The authors should consider using a more intuitive abbreviation like apoB-LPs or something similar. It is only a little longer, but much clearer.
- 2) The conclusion that the LipoGlo signal does not disrupt normal production, secretion and turnover of lipoproteins appears based on the observation that the homozygous fish are healthy, fertile with no abnormal or morphological phenotypes. This seems like light evidence. One would expect to see some sort of size pattern analysis between WT and homozygous fish to show that they are similar. Also, a demonstration that the modified LPs can interact with the LDL receptor to a similar extent as WT would also be in order. I realize that it is difficult to get enough material for these types of validations from fish. But these days it should be relatively straightforward to produce this same tag in a mouse system where the consequences of the tag attachment can be very easily studied and compared to non-labeled (i.e. LDL-R binding, size pattern, etc). As it stands, I am not convinced by the data presented that the tag is completely benign for apoB function.
- 3) The method uses crude larval homogenates. Presumably, this captures those lipoproteins in plasma. However, with cellular disruption that is also undoubtedly occurring, how can the authors distinguish between mature plasma particles and those intracellular particles still undergoing synthesis?
- 4) The size analyses of the homogenates seem problematic. The authors used a Di-I labeled (presumably human) LDL as a standard in the analysis. However, the migration of this standard in Fig. 3a clearly falls between IDL and VLDL in the size profile laid out in Fig. 3b. For example, the standard runs just below the particles classified as "VLDL" in the apoC2 KO homogenates. The authors state that the Di-I stain reduces the mobility of the standard (data not shown). So why use this as a standard? It would seem much more rigorous to introduce this same tag into mouse apoB 100 (as suggested above) and use mouse plasma lipoproteins (identically tagged) as a proper and much more comprehensive reference.
- 5) The subclass abundance plots do not make sense in Fig. 3F. Take the first panel, WT. The plot suggests that the levels of VLDL at day 3 are lower than they are at day 5. This is inconsistent with the gel image in panel C. Another example is apparent in the apoC2^{-/-} plot. The graph suggests that VLDL levels in days 3 and 4 are on par with days 5 and 6. However, they are clearly reduced in day 5 and 6 in the gel image.
- 6) Some figure references in the text are incorrect. Example, line 293.

7) The authors have not directly shown that the NanoLuc label always remains associated with the lipoprotein. This is mentioned in the discussion, but it is important. If there were significant cleavage of the tag occurring then the resulting data would no longer reflect the apoB-LP. Again, validation studies in the mouse, where there is enough plasma to actually track the lipoproteins themselves, would seem to be a needed step in the validation of this system.

8) The identification of pla2g12b is interesting and does suggest a high throughput utility for the system. However, I was surprised that if several mutant lines from the zebrafish mutation project that “had predicted mutations in genes involved in lipid metabolic pathways” were indeed studied, why was there only 1 hit? What were the other putative knock outs/mutations tested? Have those been shown to affect apoB-LPs in mice or humans? If so, why were they missed in the zebrafish? This information would go directly to the sensitivity of this assay in a high throughput setting.

Introduction to response:

We would like to thank the reviewers for their thorough evaluation of our manuscript entitled "LipoGlo: A sensitive and specific reporter of atherogenic lipoproteins". We are pleased that the reviewers appreciate the novelty and potential significance of this approach in studying the biology of ApoB-containing lipoproteins, especially as it relates to high-throughput genetic and small-molecule screening. The reviewers also highlighted several major concerns, and in response we have performed numerous additional experiments and significantly revised the manuscript as outlined below

Three major concerns stood out in the review as particularly essential to the rigor of this manuscript. Firstly, reviewers expressed concerns that the NanoLuc tag may not remain physically attached to ApoB. To address this, we performed denaturing polyacrylamide gel electrophoresis (SDS-PAGE) and western blotting and detected a prominent single band corresponding to the high molecular-weight ApoB-NanoLuc fusion protein, indicating that essentially all of the NanoLuc protein remains attached to ApoB. Secondly, reviewers expressed concern that NanoLuc may disrupt the lipoprotein profile. We add significant discussion of how existing experiments support that the lipoprotein profile remains unchanged, and performed additional experiments using lipophilic dyes to stain adult plasma and show that the profile in transgenic animals is indistinguishable from wild-type. Lastly, reviewers expressed concern that the lipoprotein localization may be altered as a result of the NanoLuc tag, as there is little evidence to corroborate the novel observation of lipoproteins associated with the myosepta and central nervous system. We addressed this concern using experiments suggested by reviewers, and have developed a new set of methods that uses Dil or Dil-labeled LDL as an orthogonal approach to track lipoproteins *in vivo*. This alternative approach corroborated our earlier conclusions, which we have further strengthened with additional citations from the literature.

Here we seek to establish a proof-of-principle that tagged lipoproteins serve as a powerful new strategy to advance our understanding of lipoprotein biology, particularly using the zebrafish system, and provide an example of how this can be achieved and validated. Publication of this manuscript in its current form can be used to guide the simultaneous translation of this approach to numerous additional model systems, including mouse.

The remaining concerns are addressed on a point-by-point basis below, and we have attached the significantly revised manuscript to be reconsidered for publication.

Response to specific concerns:

Reviewer #1:

(Remarks to the Author):

The authors develop a new method to track and image distribution of ApoB-containing lipoproteins based on a genetic fusion with the luciferase enzyme, NanoLuc. With this genetic reporter, they present assays for analyzing total amount of ApoB lipoproteins, particle types based on electrophoresis, and distribution in vivo based on imaging fixed zebrafish. They establish the approach can be applied to testing mutants and compounds. While NanoLuc has been fused to other proteins and used for imaging studies in vitro and in vivo, the described application is novel and applicable to relevant biologic questions. The presented in vivo images are convincing. Statistics and methods are appropriate to allow investigators to reproduce the imaging methods. Specific comments are listed below:

Reviewer #1 Point 1: In the discussion, the investigators state that NanoLuc predominantly remains attached to ApoB. This statement needs quantitative support and documentation as integrity of the fusion protein in all assays and in vivo is critical to interpretation of data.

Two existing lines of evidence previously included in the paper provide quantitative support that the light we detect is emitted from an intact ApoB-NanoLuc fusion protein. However, the Reviewer challenged us to provide some additional evidence to further support this point. In the first version of the manuscript we described experiments using native-PAGE followed by in-gel NanoLuc detection, finding that essentially all NanoLuc signal appears in the ZM, VLDL, IDL, and LDL regions of the gel. These data are consistent with NanoLuc remaining attached to ApoB-containing lipoproteins. When protease activity cleaves the bond between ApoB and NanoLuc, stereotypic degradation products are visible in the lower portion of the native gel where free proteins would migrate as shown in Supplementary Figure 6. Secondly, density gradient ultracentrifugation showed that NanoLuc activity was essentially undetectable in the highest density fraction (Fraction 1, Figure 4b), which corresponds to the density of free proteins (~1.35 g/mL). This indicates that the NanoLuc protein is almost exclusively associated with ApoB-containing lipoproteins. In this revised manuscript we have added an additional experiment using denaturing polyacrylamide gel electrophoresis (SDS-PAGE) to show that a specific NanoLuc antibody detects a protein migrating as a very high molecular weight entity (>250 kDa) which co-migrates with ApoB derived from human LDL, conclusively showing that NanoLuc remains fused to Apolipoprotein-B. No smaller molecular weight products were detectable, indicating that essentially all NanoLuc protein produced remains attached to ApoB. These data have been added as a new panel to Supplemental Figure 2d. Discussion of this additional experiment has been added to the main manuscript (as well as pasted below) and the protocol is described in detail in the “Western Blotting” section added to materials and methods (also pasted below).

(Results)

Denaturing polyacrylamide gel electrophoresis (SDS-PAGE) followed by labeling with specific ApoB and NanoLuc antibodies reveal a single high molecular-weight band (>250 kDa) that corresponds with the expected migration pattern of an ApoB-NanoLuc fusion protein. Degradation products and/or free-NanoLuc protein with lower molecular weights were undetectable (Supplementary Fig. 2d).. Together, these data indicate that the LipoGlo reporter signal is directly proportional to ApoB levels.

(Methods)

Western Blotting

Protein extraction was performed on 10 pooled 3 dpf larvae per sample. Larvae were transferred to a 1.5 mL microcentrifuge tube, excess liquid was removed, and 100 μ L of RIPA buffer containing 3x protease inhibitor cocktail was added. Larvae were immediately homogenized using a pellet pestle, and incubated at 4°C for 15 minutes with shaking. Samples were then centrifuged at 12,000 rcf for 5 minutes and the

supernatant was mixed with an equal volume of 2x Laemmli buffer (BioRad, 1610737) and heated to 95°C for 5 minutes in a thermal cycler. Dil-LDL (L3482, Thermofisher Scientific) was diluted 100-fold in RIPA buffer and extracted as above to be used as an indicator of the migration pattern of APOB, and Halo-Tagged NanoLuc protein (Promega, CS188401) was diluted 10,000-fold in RIPA buffer and used as an indicator of the migration of free NanoLuc protein. Precision Plus Protein All Blue Prestained Protein Standards (BioRad, 1610373) was used as a molecular weight marker.

25 µL of the resulting sample was loaded onto a precast 4-20% gradient gel (BioRad, 4561093) and separated at 70 V for 30 minutes and 90 V for 60 minutes. Proteins were then transferred to a PVDF membrane with the Trans-blot Turbo Transfer System (BioRad, 1704150) using a custom transfer program optimized to ensure transfer of high-molecular weight proteins (1.3 Amp constant for 15 minutes). The blot was blocked in 5% milk for 1 hour, and then probed simultaneously with primary antibodies binding NanoLuc (R&D Systems, MAB10026-100, 1:200 dilution) and human APOB (Meridian Life Sciences, K45253G, 1:200 dilution) for 4 hours at room temperature in 2.5% milk. The blot was then rinsed 3 times for 5 minutes each in TBST, and probed with fluorescent secondary antibodies (LICOR Biosciences, IRDye 800CW Donkey anti-Goat IgG, 925-32214, and IRDye 680RD Donkey anti-Mouse IgG, 925-68072, 1:5,000 dilution) for 1 hour at room temperature. The blot was then rinsed as above and imaged in the 700 and 800 nm channels for 2 minutes each using the Odyssey Fc (LI-COR Biosciences)

Reviewer #1 Point 2: The authors should clarify the magnitude of background signal in assays relative to signal produced by NanoLuc-ApoB. Did the authors subtract background values from presented data in graphs?

The Reviewer makes a good point that providing signal to noise/background ratio data and a discussion of its relevance is important and should be included in the manuscript. To address this point we have added a new panel (Supplementary Figure 7i). Discussion of the signal to noise ratio data and method of blank subtraction was added to the quantification and statistical analysis section of the methods (pasted below).

(Methods)

One of the strengths of a chemiluminescent reporter is that it has excellent signal to background ratios. In our hands, the signal to background ratio varies significantly between assays, reaching approximately 30,000:1 in plate-based assays, 30:1 in gel-based assays, and 13:1 in microscopy assays (Supplementary Fig. 7i). Blank subtraction was therefore not performed for the above analyses, as it was found not to have an impact on the results. This is likely due to the fact that background signal is negligible in the plate-based assays, and the remaining assays not only have very low levels of background signal, but are also quantified in terms of relative (rather than absolute) signal and thus less susceptible to skewing from the background signal.

Reviewer #1 Point 3: The authors should comment on stability of the NanoLuc signal over time in various assays as this is important for reliable quantification.

We agree with the Reviewer that the stability of NanoLuc signal over time is important for reliable quantification of the plate read data, and this information is provided in Supplementary Figure 7d showing remarkable stability of the NanoLuc signal over the course of 20 minutes across a broad concentration series. We apologize for omission of kinetic data for the electrophoresis and microscopy assays, which has now been quantified and added (Supplementary Fig. 7e – 7h). Importantly, these data demonstrate that while the absolute intensity of NanoLuc signal does vary over time for the LipoGlo-electrophoresis and LipoGlo-microscopy techniques, the information quantified (the relative migration and localization patterns) are quite robust to moderate fluctuations in incubation time. Discussion of these findings has been added to the methods section of the manuscript (also pasted below).

(Methods)

Effort should be made to ensure a consistent equilibration time between substrate addition and gel imaging. While this assay is relatively robust to small variations in incubation time (Supplementary Fig. 7g-7h), equilibration times below 5 minutes or greater than 20 minutes are not recommended.

While the absolute intensity of NanoLuc signal will decay gradually over time (Supplementary Fig. 7e-7f), the distribution of NanoLuc signal is relatively robust to slight variation in incubation time. Nonetheless, it is recommended that larvae be mounted and imaged with as consistent of an incubation period as possible (ideally less than 30 minutes).

Reviewer #2:

(Remarks to the Author):

This manuscript from Thierer et al., described the generation of a new knockin zebrafish in which the apoB protein is tagged with luciferase (NanoLuc). The authors showed that this novel genetically modified animal can be used to track some of metabolic features of apo-B containing lipoproteins in a spatiotemporal manner, using *mtp* or *apoc2* loss-of-function animals or using a MTP inhibitor. Indeed, few, if there is any previous technique can track the size of apoB lipoproteins in vivo using live animals. This platform can also be used to screen compounds interfering with lipoprotein metabolism in a large scale and be adopted to other organisms. Furthermore, leveraging this novel animal model, they defined the new function of *pla2g12b* as a critical regulator of apoB-containing lipoprotein size. In general, this is a very interesting paper demonstrates the powerful usage of zebrafish animal model in combination with versatile gene targeting. The manuscript was well-written. While this is a well-conducted study, a few improvements may further strength their findings.

Major concerns:

Reviewer #2 Point 1: As far as we know, most lipoproteins reside in the vasculature. The in vivo luciferase signal did not show enrichment in the blood vessels (Figure. 5a-d) but instead in the brain, spinal cord, liver and intestine. What would be the in vivo distribution of apoB-lipoproteins without the luciferase tag? The author may inject human fluorescently labeled human LDL and monitor their distribution.

At the onset of this project, the authors had a similar perspective to Reviewer 2 in that “most lipoproteins reside in the vasculature”. However, based on our extensive review of the literature and new data provided in this paper, we believe that this perspective is biased by how lipoproteins have been traditionally studied. In general, plasma is used as the starting material for the study of lipoproteins, and thus any lipoproteins outside of the plasma are almost never examined. In this paper, we highlight an abundance of lipoproteins in several extravascular tissues including the liver, intestine, yolk-syncytial layer (YSL), central nervous system (CNS), and myosepta. While we were initially surprised by some of these observations, upon further investigation we found significant support for each of these observations in the literature. Firstly, lipoproteins are produced by the liver, intestine, and YSL, and the LipoGlo signal in these organs likely reflects both the nascent lipoproteins being secreted by these tissues, and in the case of the liver, lipoproteins being endocytosed from the circulation. Lipoproteins have also been previously detected in embryonic cerebrospinal fluid of chick embryos (see added discussion below), thus validating our detection of LipoGlo signal associated with the CNS. Lastly, individuals with mutations causing abnormally high levels of LDL (homozygous familial hypercholesterolemia) develop severe tendon defects, which is consistent with the localization of LipoGlo signal in the tendinous myosepta of larval zebrafish. Additional discussion and the relevant citations have been added to the manuscript and pasted below.

We would like to thank the reviewer for proposing a powerful experiment to validate that the observed localization pattern is not an artifact of the LipoGlo system by injecting fluorescently labelled LDL into the bloodstream. We have not only performed this experiment, but performed a related experiment using injection of Dil (a lipophilic fluorescent dye) into the larval yolk to fluorescently label endogenous zebrafish lipoproteins. The results of both of these experiments corroborate the findings of the LipoGlo-Microscopy experiment, confirming that the localization patterns observed are not an artifact of the NanoLuc tag. These findings are presented in Supplementary Fig. 5c-5e, and discussed in the main text in new sections of the methods, results, and discussion (pasted below).

(Methods) Fluorescent labeling and imaging of ApoB-LPs

For injection of Human Dil-labelled LDL into zebrafish larvae, larvae were first anesthetized and mounted laterally in a petri dish in a 50 μ L droplet of 1 % low-melt agarose prepared in zebrafish embryo medium. After the agarose had solidified, forceps were used to remove a small portion of agarose just dorsal to the anterior portion of the yolk, providing an access window for injection. The petri dish was then filled with embryo medium to ensure that larvae did not dry out during injection and imaging. Human Dil-labelled LDL (L3482, ThermoFisher Scientific) was then loaded into an microinjection needle and calibrated to an injection volume of 4 nL. Larvae were then injected into the common cardinal vein (CCV) through the agarose-free access window, or injected directly into the yolk. Larvae were imaged within 1 hour of injection, and then carefully liberated from the agarose using fine forceps.

For injection of Dil into the larval yolk, Dil (D282, ThermoFisher Scientific) was resuspended to 30 mg/mL in DMSO. This solution was then loaded into an injection needle and calibrated to 4 nL injection volume. 1 dpf was selected as an ideal time point for injection because at this stage the yolk is fully segregated from the embryo, yet the embryos are still within their chorions which eliminates the necessity for mounting in agarose prior to injection.

Larvae were then imaged on an SMZ25 microscope equipped with a Cy3 filter cube. Larvae were imaged after being mounted in either 1% low-melt agarose (in the case of 2 dpf injections), or were anesthetized and imaged in 3% methylcellulose. Image brightness was adjusted arbitrarily in the display to ensure visibility of relevant structures.

(Results) Dil-labelled LDL confirms the localization patterns observed with LipoGlo-Microscopy

ApoB-LPs have primarily been studied for their roles in the circulatory system, where they transport lipid between tissues and also contribute to the progression of atherosclerosis. However, LipoGlo-Microscopy experiments revealed two highly unexpected patterns of lipoprotein localization. Firstly, fasted larvae retain high levels of APOB-LPs associated with the central nervous system. Secondly, a high level of ApoB-LPs appears in a chevron pattern along the trunk of zebrafish larvae, which corresponds to the myosepta, the tendinous tissue connecting body segments. To validate that this localization pattern was not an artifact resulting from the introduction of the NanoLuc reporter, we developed two orthogonal approaches to monitor the localization of ApoB-LPs in zebrafish larvae using a fluorescent lipophilic dye (Dil).

Dil has frequently been used to label lipoprotein particles, as its spectral properties change dramatically when it is incorporated into a phospholipid monolayer, thus reducing background fluorescence from unincorporated dye. As a means of visualizing LDL localization in vivo, commercially available human Dil-LDL was injected into the zebrafish bloodstream at 2 dpf and then imaged at various time points throughout development (Supplementary Fig. 5c). Immediately following injection (2 dpf), bright Dil fluorescence was readily detectable throughout the vascular system, which is particularly clear in the tail vasculature including the caudal artery (CA), caudal vein plexus (CVP), and the intersegmental vessels (ISV). However, imaging at later time points (4 and 6 dpf) revealed significant accumulation in myosepta (MS) and the spinal cord (SC), closely mirroring the localization pattern observed in LipoGlo microscopy. However, in contrast to the LipoGlo microscopy experiments, significant signal accumulated in bright puncta in the ventral posterior of the trunk, which most likely corresponds to macrophages in the caudal hematopoietic tissue (CHT). This result indicates that human Dil-LDL may be immunogenic, either because it is not derived from zebrafish or it has become oxidized or aggregated during storage.

As a negative control, human Dil-LDL was also injected into the yolk of zebrafish larvae (Supplementary Fig. 5d). Immediately after injection, signal was essentially undetectable outside of the yolk, confirming that it has not reached the vasculature. However, approximately 50% of larvae injected into the yolk accrued significant signal outside of the yolk by 6 dpf, where it appeared to mark similar structures as seen in the previous experiment, although signal in the CHT appeared less pronounced. This observation suggests that Dil injected into the yolk (even in the form of human Dil-LDL) could be transferred to endogenous lipoproteins and secreted.

To test whether Dil could be used to monitor endogenous APOB-LPs, we injected Dil directly into the yolk of zebrafish larvae at 1 dpf (Supplementary Fig. 5e). Dil signal closely mirrored the LipoGlo microscopy experiments throughout development. Importantly, this Dil-labeling paradigm showed clear enrichment in the spinal cord and myosepta by 6 dpf, validating the findings of the LipoGlo microscopy experiment.

(Discussion)

Previous work has shown that LDL is present in the embryonic cerebrospinal fluid of chick embryos. There LDL can interact with SCO-spondin protein that contains multiple LDL binding domains and forms the Reissner fiber, a protein fiber found in the central canal of all chordates [51]. In zebrafish embryos this fiber contributes to proper body axis formation [52].

...

We also developed an orthogonal approach to monitor APOB-LP localization in vivo using Dil, a fluorescent lipophilic dye routinely used to label lipoproteins. Both human and endogenous lipoproteins labeled with Dil can be monitored in vivo, and show effectively identical localization patterns to lipoproteins labeled with NanoLuc, thus validating the localization patterns observed with LipoGlo microscopy.

Reviewer #2 Point 2: What is the lipoprotein profile of the adult LipoGlo animal compared to unmodified AB animals, in the presence and absence of hyperlipidemia? Such as the LDL-C and HDL-C as measured in Ohare et al., J Lipid Res. 2014 Nov;55(11):2242-53

We would like to thank the reviewer for highlighting this important point. Although we perform extensive characterization of the larval lipoprotein profile using LipoGlo, since there are no alternative techniques to characterize the lipoprotein profile without the lipoGlo reporter, we cannot be certain that LipoGlo does not induce changes in the larval lipoprotein profile. To address this issue, we have performed additional experiments to characterize the lipoprotein profile of adult fish in both the wild-type and LipoGlo backgrounds using Di-I staining and native-PAGE. We found these profiles to be indistinguishable. This experiment has been added as Supplementary Fig. 8, and corresponding sections have been added to the methods, results, and discussion (pasted below). Unfortunately we did not have a sufficient number of adult fish of the appropriate genotypes to perform these experiments in hyper/hypo-lipidemic mutants, these experiments are on-going and are beyond the scope of the current manuscript.

(Methods) Dil labeling of adult zebrafish plasma

Adult zebrafish were anesthetized and transferred to a dissection stage covered with a kimwipe moistened with anesthetic diluted in system water. Tails were then resected at the anterior of the anal fin to expose the caudal artery. An EDTA-coated capillary fitted on the end of a p20 pipette tip was used to slowly extract whole blood from the artery, and the fish was immediately euthanized in an ice bath following extraction. Blood was transferred to a 1.5 mL microcentrifuge tube and spun at 6,000 rcf for 5 minutes to pellet blood cells, and the resulting plasma was transferred to a new tube. The Plasma was then diluted ten-fold in a solution of APOB-LP stabilization buffer containing 30 mg/mL Dil. This mixture was incubated at 37°C for two hours, and stored at 10°C overnight or used immediately for electrophoresis. 5 µL of the resulting stained lipoproteins were mixed with 45 µL of APOB-LP buffer containing loading dye and loaded onto a 3% native polyacrylamide gel. Native-PAGE and gel imaging were performed as described above for LipoGlo-electrophoresis.

(Results) Adult zebrafish plasma labeled with Dil confirms that LipoGlo does not disrupt the APOB-LP profile

LipoGlo has revealed numerous aspects of the lipoprotein profile in zebrafish larvae, many of which are in line with our current understanding of lipoprotein homeostasis. However, as no alternative methods exist to study the lipoprotein profile in zebrafish larvae at this level of sensitivity or resolution with regard to particle size and number, it was not possible to compare the lipoprotein profile between wild-type and LipoGlo larval individuals. However, using plasma extracted from adult animals the hypothesis that LipoGlo labeling does not alter the plasma lipoprotein profile was tested. Adult zebrafish plasma lipoprotein profiles determined using native-PAGE from WT animals that were labeled with Dil were essentially indistinguishable from those

homozygous for the LipoGlo reporter, although there was significant variation between individuals (likely as a result of variations in activity and feeding behavior) (Supplementary Fig. 8). These data indicate that the addition of NanoLuc to the carboxy-terminal of ApoB does not disrupt the lipoprotein profile.

While no differences were apparent between wild-type and LipoGlo animals, we were able to detect a significant Dil-positive (NanoLuc-negative) band that is present exclusively in female plasma, and absent in males. This most likely corresponds to vitellogenin, a large protein used to shuttle lipids through the bloodstream that will eventually be used in egg production. While several additional bands are present in the Dil-stained plasma, without additional molecular markers it is difficult to conclusively determine what these species may be, but high-density lipoproteins would also be expected to stain with Dil as well.

(Discussion) The LipoGlo reporter does not disrupt lipoprotein homeostasis

When generating a fusion protein, it is essential to evaluate whether introduction of the tag disrupts native protein function. This is particularly important in the case of tagged lipoproteins, as these particles have a complex life cycle that involves interaction with numerous cell and tissue types. The fact that fish homozygous for the LipoGlo reporter are viable, fertile, and free from any overt morphological defects served as encouraging preliminary evidence that metabolism was not greatly disrupted. To evaluate whether the NanoLuc tag disrupted lipoprotein homeostasis in a more subtle way, LipoGlo larvae were subjected to various genetic, dietary, and pharmacological manipulations known to affect the lipoprotein profile. These results validated that NanoLuc-tagged lipoproteins exhibit all of the central hallmarks of endogenous APOB-LPs, including MTP-dependent maturation, APOC2-dependant lipolysis, responsiveness to nutrient availability, and expected density and size distributions.

Further, LipoGlo microscopy showed that APOB-LPs are initially only detectable in lipoprotein-producing tissues, and then distribute to peripheral tissues, and finally become undetectable in peripheral tissues when larvae are fasted. These observations are constant with tagged lipoproteins being secreted into the circulatory system, processed in the peripheral circulation, and eventually endocytosed, as would be expected from untagged lipoproteins. The Dil stained plasma lipoprotein profiles from adult zebrafish homozygous for the NanoLuc reporter and wild-type controls were indistinguishable. Taken together, these observations indicate that the carboxy-terminal fusion of NanoLuc to ApoBb.1 does not detectably alter lipoprotein homeostasis. This finding establishes a precedent for the use of ApoB-fusion proteins as a sensitive and specific approach to monitor atherogenic lipoproteins which will likely be generalizable to additional model systems. This approach can also be expanded to use alternative tags such as fluorescent reporters for high-resolution imaging or affinity tags to study the lipoprotein interactome. Potential applications of LipoGlo thus extend well beyond the study of lipoprotein abundance, size, and localization using zebrafish.

Reviewer #2 Point 3: Figures 3 and 4 show differential apoB lipoprotein fractions using native gel electrophoresis and ultracentrifugation-based isolation. A verification of the presence of cholesterol using NBD cholesterol staining (or TopFLuo cholesterol-containing food feeding) and apoB using WB can further strength the conclusions.

We fully agree that the verification of the presence of ApoB or cholesterol would strengthen the support for these conclusions. Unfortunately, several technical hurdles complicate the execution of experiments. Firstly, we have not yet identified an antibody suitable for detection of ApoB in zebrafish (demonstrated in Supplementary Figure 2d). However, the response to Reviewer # 1 point 1 (above) summarizes the additional experiments we have performed to validate that NanoLuc remains attached to ApoB, and we are unable to detect any NanoLuc that is not associated with ApoB, verifying that light emission from NanoLuc is a valid method to quantify ApoB levels.

The reviewer recommends an experiment involving top-Fluor cholesterol feeding to validate the presence of cholesterol in the native gel and density fractions. While we agree that this would be a valuable experiment in principle, we have serious concerns that this fluorescent marker would not be sufficiently sensitive to detect the lipoproteins derived from 1/10th of an individual larva. These concerns are fueled by the observation that fluorescent reporters have significantly higher background than chemiluminescent reporters such as NanoLuc, as can be observed in Supplementary Figure 3. Here, human LDL is saturated with the fluorescent dye Dil. When diluted to slightly over 100-fold to ~.01 mg/mL (or ~20 μ M), the signal becomes nearly undetectable. By contrast, we estimate that the larval homogenate contains LDL at a concentration of approximately 1 nM, or 20,000 times below the detection limit of Di-I LDL, effectively precluding the use of

fluorescent lipids to trace lipoproteins in larval homogenate. However, we were able to obtain concentrated lipoproteins from adult plasma and label them with Dil, a lipophilic dye routinely used to label lipoproteins. These results are presented as a new Supplementary Figure (Supplementary Figure 8), and although Dil appears to label numerous additional species in zebrafish plasma (including vitellogenin), there is excellent concordance between the LipoGlo signal and Dil (as discussed above).

We would also like to direct the reviewer to a publication referenced in the manuscript (Liu, C., et al., *Apoc2 loss-of-function zebrafish mutant as a genetic model of hyperlipidemia*. *Dis Model Mech*, 2015), which pools together plasma from 40 adult zebrafish and is able to quantify the cholesterol and triglycerides in the corresponding lipoprotein fractions.

We have also validated the lipoprotein fractions in numerous other ways throughout the manuscript, for example by demonstrating that these fractions (i) respond to dietary lipid availability, (ii) respond to ApoB-lowering pharmaceuticals, (iii) have the expected density of lipoproteins, (iv) show the proper size and morphology when directly visualized by electron microscopy, and (v) depend on the canonical cofactors MTP and ApoC2 for their production, turnover, and overall size distribution profile. Taken together, these observations provide a preponderance of evidence to support that these different fractions are indeed different classes/sizes of ApoB-containing lipoproteins.

Reviewer #2 Point 4: The author identified the novel pla2g12b gene in modifying lipoprotein size. What is the current study on this gene? Any association of this gene with cardiovascular disease and lipoprotein metabolism? The author may test if indeed pla2g12b deficiency augments foam cell formation in apoC2 mutant or control animals.

We share the reviewer's curiosity surrounding the Pla2g12b gene. This gene is very poorly understood, and the little data available essentially shows that this protein is catalytically inactive to phospholipid substrates, yet somehow promotes efficient triglyceride secretion via ApoB-containing lipoproteins. However, no mechanism has been proposed to explain this activity. We are very curious to see what impact this protein has on the incidence of cardiovascular disease, but no link has been reported previously. This is a particularly interesting case where abnormally small lipoproteins are produced but there is a smaller overall number of particles, which would be expected to have counteracting effects on cardiovascular disease risk and thus may explain a lack of previously published associations between this gene and cardiovascular disease. However, we believe that the detailed study of this protein (such as its mechanism of action and its impact on cardiovascular disease risk) are beyond the scope of this study. Rather, in the context of this study, the identification of Pla2g12b as a novel regulator of lipoprotein size serves as an illustrative example of the utility of the LipoGlo system in discovering new regulators of lipoprotein size, abundance, and localization. We hope that the discovery of additional genes such as this one will provide researchers with a wide array of mutations that disrupt the lipoprotein biology, all of which can be tested for their impact on cardiovascular disease risk.

Reviewer #2 Point 5 (minor): It is interesting that the animals were PFA fixed first and then the luciferase substrate was added. Is it because of high sensitivity of this NanoLuc?

PFA and Tween-20 were added to facilitate permeabilization of the zebrafish larvae. Unlike luciferin (the substrate of firefly luciferase) which freely diffuses into living cells and tissues, the NanoLuc substrate furimazine does not freely diffuse into living tissues. The remarkable stability of the NanoLuc enzyme allows it to retain its activity despite mild fixation with PFA.

Reviewer #2 Point 6 (minor): What changes in the apoB containing lipoproteins when their size become smaller and render them more atherogenic? Any in vitro or in vivo evidence?

We apologize for omission of this interesting topic. A statement and critical reference has been added to address this point (pasted below).

(Introduction)

The higher atherogenic potential of small dense LDL particles (sdLDL) has been attributed to a combination of three properties [7], including increased rates of intimal invasion, reduced receptor-mediated clearance, and increased susceptibility to oxidation.

Reviewer #3:

(Remarks to the Author):

This manuscript describes experiments to develop and validate the LipoGlo reporter for apoB containing lipoproteins in zebrafish. The authors have tagged the zebrafish homolog of apoB with a luciferase tag which allows the tracking of its lipoproteins around the transparent fish larvae. In addition, the larvae can be ground up and the lipoproteins analyzed by electrophoretic techniques to determine general distributions between CM, VLDL, IDL and LDL. The authors used the system to identify potentially new sites of apoB-LP association within the larval body and, as a proof of concept, identified a gene that likely affects apoB-LP metabolism in fish. Overall, this is an interesting system. The resolution of the system, both in terms of the gel electrophoresis size analysis and the whole body localization studies, is perhaps a bit disappointing. I was hoping that it might be possible to see associations that are more related to atherosclerosis development like accumulation of apoB-LPs in the vasculature, for example. However, it clearly has potential uses in screening for genes and drug treatments that affect apoB-LP metabolism. However, there were some deficiencies in the characterization of the system and inconsistencies in the data that reduced enthusiasm for this work in its current state. My specific comments are outlined below.

Reviewer #3 Point 1: The abbreviation ABCLs seems a bit clunky for apoB containing lipoproteins. The authors should consider using a more intuitive abbreviation like apoB-LPs or something similar. It is only a little longer, but much clearer.

The authors were also frustrated by the imprecise and inconsistent terminology used to discuss ApoB-containing lipoproteins in the existing literature. At the Reviewer's recommendation, we have adopted the suggested acronym ApoB-LPs and incorporated it throughout the manuscript.

Reviewer #3 Point 2: The conclusion that the LipoGlo signal does not disrupt normal production, secretion and turnover of lipoproteins appears based on the observation that the homozygous fish are healthy, fertile with no abnormal or morphological phenotypes. This seems like light evidence. One would expect to see some sort of size pattern analysis between WT and homozygous fish to show that they are similar. Also, a demonstration that the modified LPs can interact with the LDL receptor to a similar extent as WT would also be in order. I realize that it is difficult to get enough material for these types of validations from fish. But these days it should be relatively straightforward to produce this same tag in a mouse system where the consequences of the tag attachment can be very easily studied and compared to non-labeled (i.e. LDL-R binding, size pattern, etc). As it stands, I am not convinced by the data presented that the tag is completely benign for apoB function.

We agree with the reviewer that the lack of an overt morphological phenotype would constitute very light evidence that lipoprotein homeostasis is not disrupted. However, throughout the paper we have assembled numerous lines of evidence to validate that various aspects of lipoprotein homeostasis are preserved in LipoGlo animals. For example, the fasting and feeding experiments demonstrate that lipoprotein production is lipid-dependent, and fasting leads to lipoprotein uptake and turnover. Studies in mutant lines show that lipoprotein production is MTP dependent, and that particle turnover is ApoC2 dependent, thus validating that interaction with these central lipoprotein processing genes is intact in LipoGlo larvae. While interaction with the LDL-receptor is not tested directly, the drop in total NanoLuc signal and clearance from peripheral tissues as wild-type larvae are fasted implies that receptor mediated endocytosis is taking place. Biochemical experiments show that the particles are the appropriate size and density, and have the appropriate morphology on electron micrographs.

We concede that despite all the above evidence, a critical experiment is missing. Given that it is not possible to characterize the larval lipoprotein profile without using LipoGlo, we were unable to compare the lipoprotein profiles between wild-type and LipoGlo larvae. The reviewer suggests recreating the LipoGlo reporter in mouse, which we believe is an exciting future direction but is beyond the scope of this study. As an alternative, we have compared the adult plasma lipoprotein profiles between wild-type and LipoGlo fish, as described above in the response to Reviewer #2 point 2. This experiment (added as Supplementary Figure 8) shows that the adult plasma profile is indistinguishable between wild-type and LipoGlo animals.

We have also added another experiment to strengthen this conclusion, which was also performed in response to Reviewer #2 point 1. We found that when DiI is used to label endogenous zebrafish lipoproteins (without the NanoLuc tag), the same localization patterns are observed when we compare to the results of the LipoGlo microscopy experiments. This provides further support that lipoproteins tagged with NanoLuc does not disrupt production, secretion, or turnover of lipoproteins. These topics are discussed throughout the manuscript as outlined in the responses to Reviewer #2 points 1 and 2.

Reviewer #3 Point 3: The method uses crude larval homogenates. Presumably, this captures those lipoproteins in plasma. However, with cellular disruption that is also undoubtedly occurring, how can the authors distinguish between mature plasma particles and those intracellular particles still undergoing synthesis?

The reviewer is correct in noting that this method uses crude homogenates, and thus includes nascent intracellular lipoproteins that would not be included in a typical analysis of plasma lipoproteins. While we cannot conclusively distinguish between particles in the plasma versus those undergoing synthesis, we have made several observations that help to address this point.

Our working hypothesis is that nascent lipoproteins are still contained within or associated with components of the secretory pathway, including the ER and pre-VLDL transport vesicles, which greatly retards their mobility. We thus suspect that all intracellular ApoB is present within the zero-mobility (ZM) band. This is evidenced by the fact that small lipoproteins are essentially undetectable in ApoC2 mutants (Figure 3c). If nascent lipoproteins that had not yet been fully lipidated and secreted were able to migrate into the gel, we would be able to detect these small lipoproteins migrating far into the gel, but no such particles are observed and thus are presumably retained in the ZM band. As further support for this model, in 6 dpf *mtp*^{-/-} mutant larvae, essentially all lipoproteins are present in the lipoprotein-producing tissues (liver and intestine, Figure 5d). Correspondingly, essentially all signal on the native gel is restricted to the ZM band. Thus, while there is ambiguity within the ZM band (relevant section pasted below), we are confident that lipoproteins migrating into the gel are indeed plasma lipoproteins.

(Results)

ApoB-LPs that remain within the loading well are classified as the “zero mobility” (ZM) fraction, which should include chylomicrons [37], remnants, aggregates [38], and intracellular ApoB complexed with components of the secretory pathway (such as the ER, golgi, and other secretory vesicles) [39].

Reviewer #3 Point 4: The size analyses of the homogenates seem problematic. The authors used a Di-I labeled (presumably human) LDL as a standard in the analysis. However, the migration of this standard in Fig. 3a clearly falls between IDL and VLDL in the size profile laid out in Fig. 3b. For example, the standard runs just below the particles classified as “VLDL” in the apoC2 KO homogenates. The authors state that the Di-I stain reduces the mobility of the standard (data not shown). So why use this as a standard? It would seem much more rigorous to introduce this same tag into mouse apoB 100 (as suggested above) and use mouse plasma lipoproteins (identically tagged) as a proper and much more comprehensive reference.

Our goal for developing an appropriate migration standard was to increase reproducibility and ensure consistent results between labs. We believe that Human DiI-LDL is the ideal solution to achieve these goals, primarily because it is commercially available. This enables labs around the world to access a consistent product that meets industry-level quality control, provides customer support, can be purchased and shipped

rapidly, and can meet essentially any level of demand. While we are enthusiastic to recreate the LipoGlo reporter in a mouse system and those efforts are beginning in the Farber lab and with collaborators, it is outside of the scope of the present study. Also, the use of NanoLuc-labelled mouse LDL as a normalization standard would shift the burden of production, quality control, billing, and shipping of this essential resource onto an academic entity with limited resources that would inevitably be less efficient, smaller-scale, and less reproducible than ThermoFisher Scientific (the current supplier). Supplementary Figure 3 provides detailed explanation of the utility of Dil LDL as a normalization standard, showing that it greatly reduces variability between runs despite the fact that it does not migrate at the same rate as NanoLuc-labeled zebrafish LDL. Supplementary panel 3g has also been added, which shows a dose-response between concentration of Dil and changes in electrophoretic mobility.

(Results)

Although human Dil-LDL migrates more slowly than NanoLuc-labeled LDL, which is at least partially attributable to migration retardation by Dil (Supplementary Fig. 3g), this band provides a highly reproducible standard for registration and normalization across gels (Supplementary Fig. 3).

Reviewer #3 Point 5: The subclass abundance plots do not make sense in Fig. 3F. Take the first panel, WT. The plot suggests that the levels of VLDL at day 3 are lower than they are at day 5. This is inconsistent with the gel image in panel C. Another example is apparent in the apoC2^{-/-} plot. The graph suggests that VLDL levels in days 3 and 4 are on par with days 5 and 6. However, they are clearly reduced in day 5 and 6 in the gel image.

We apologize for the confusion, and in light of the Reviewer's concerns have clarified the labelling and discussion of these plots in the main text, the labels of the Figures themselves, as well as the Figure legends. To specifically address the reviewers first example, the level of VLDL across WT development is represented by the second-lightest shade of gray in Fig. 3F (note a label was added to this panel for improved clarity). On these 100% stacked line graphs, the thickness of that shade of gray at that time point reflects the relative abundance of that species. Note that, consistent with the gel image, this species is substantially more abundant at day 3 than day 5 (this shade of gray gradually gets thinner from 2-6 dpf). These data are presented in a different way in Supplemental Fig. 4 for added clarity.

To specifically address the reviewers second example, the key distinction is that these plots represent relative abundance rather than total/absolute abundance. Indeed, there is a clear peak in signal intensity at days 3 and 4 of development, and this is fully consistent with the measurements of the total abundance of particles reported in Figure 2c (plate-reader measurements). However, even though there are changes in the total number of particles, the relative abundance of each particle class remains remarkably consistent in the ApoC2 mutant throughout development, with approximately 40% of the total LipoGlo signal in the VLDL band and 50% of the signal in the ZM band. This conclusion is also displayed in Supplemental Figure 4.

The relevant modifications to the main text are pasted below.

(Results)

Note that LipoGlo electrophoresis is only used to determine relative abundance, rather than absolute or total abundance of lipoproteins. This is useful for highlighting differences in size distributions even when the total number of APOB-LPs is vastly different. To illustrate this point, compare the ZM bands between 4 dpf and 6 dpf larvae. The ZM band is clearly significantly brighter at 4 dpf (higher absolute abundance), but it accounts for a smaller fraction of the total profile (lower relative abundance). To visualize the distribution of APOB-LP classes over time, each species was color coded with darker colors corresponding to smaller lipoproteins and plotted as an 100% stacked area chart, with the thickness of each shade corresponding to the relative abundance of that species at that time (Fig. 3f-h).

(Figure legend)

Note that relative abundance was quantified, so the sum of all species will always equal 100% despite changes in total abundance over time

Reviewer #3 Point 6: Some figure references in the text are incorrect. Example, line 293.

We thank the reviewer for calling attention to this error, and have made the appropriate corrections.

Reviewer #3 Point 7: The authors have not directly shown that the NanoLuc label always remains associated with the lipoprotein. This is mentioned in the discussion, but it is important. If there were significant cleavage of the tag occurring then the resulting data would no longer reflect the apoB-LP. Again, validation studies in the mouse, where there is enough plasma to actually track the lipoproteins themselves, would seem to be a needed step in the validation of this system.

We agree that this is a critically important point, and has been addressed with additional experiments and discussion as described above. See response to Reviewer #1 point 1.

Reviewer #3 Point 8: The identification of *pla2g12b* is interesting and does suggest a high throughput utility for the system. However, I was surprised that if several mutant lines from the zebrafish mutation project that “had predicted mutations in genes involved in lipid metabolic pathways” were indeed studied, why was there only 1 hit? What were the other putative knock outs/mutations tested? Have those been shown to affect apoB-LPs in mice or humans? If so, why were they missed in the zebrafish? This information would go directly to the sensitivity of this assay in a high throughput setting.

We apologize for the confusion, and have clarified the discussion in the main text. Briefly, the Farber lab has a longstanding relationship with the Sanger zebrafish mutation project and history of using a variety of assays to evaluate mutants from this collection for defects in lipid metabolism. However, a detailed investigation of transcript processing in several ENU alleles revealed that many of the alleles we were initially interested in testing underwent alternative splicing to compensate for their predicted mutations. Following this analysis, *Pla2g12b* and *abca1b* were the only alleles with altered transcript levels, and were thus selected as ideal candidates for breeding into the LipoGlo background. A reference was added that reports the results of the mRNA analyses (Anderson, J.L., et al., *mRNA processing in mutant zebrafish lines generated by chemical and CRISPR-mediated mutagenesis produces unexpected transcripts that escape nonsense-mediated decay*. PLoS Genet, 2017.), and the main text was modified as outlined below.

*In an effort to identify novel regulators of the ApoB-LP profile using the LipoGlo system, we analyzed a collection of mutants from the zebrafish mutation project [43] that had predicted mutations in genes involved in lipid metabolic pathways. Of the six ENU alleles studied, two alleles (*abca1b* and *pla2g12b*) were particularly promising as we detected nonsense-mediated decay in the mutant transcripts [44]. Studies of *abca1b* using LipoGlo are underway, but we have discovered that larvae homozygous for an essential splice site mutation (*sa659*) in phospholipase A2 Group XII B (*pla2g12b*) showed perturbations in their ApoB-LP profile (Fig. 6).*

REVIEWERS' COMMENTS:

Reviewer #1 (Remarks to the Author):

The revised manuscript appropriately addresses comments from the initial review.

Reviewer #2 (Remarks to the Author):

This revision from the authors has addressed this reviewer's concerns. I strongly recommend its publication in Nature Communications.

Reviewer #3 (Remarks to the Author):

N/A